

# Heavy operators and hydrodynamic tails

**Luca V. Delacrétaz**

Enrico Fermi Institute & Kadanoff Center for Theoretical Physics,
University of Chicago, Chicago, IL 60637, USA

## Abstract

The late time physics of interacting QFTs at finite temperature is controlled by hydrodynamics. For CFTs this implies that heavy operators – which are generically expected to create thermal states – can be studied semiclassically. We show that hydrodynamics universally fixes the OPE coefficients $C_{HH'L}$, on average, of all neutral light operators with two non-identical heavy ones, as a function of the scaling dimension and spin of the operators. These methods can be straightforwardly extended to CFTs with global symmetries, and generalize recent EFT results on large charge operators away from the case of minimal dimension at fixed charge. We also revisit certain aspects of late time thermal correlators in QFT and other diffusive systems.

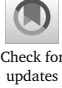

# 1 Introduction

The analytic conformal bootstrap has uncovered universal features in sparse corners of the spectrum of conformal field theories (CFTs), at large spin [1, 2] or large charge [3]. The 'middle' of the spectrum is instead exponentially dense, but reveals universal properties as well [4,5]. Some of these advances were guided by the existence of semiclassical descriptions, such as weakly interacting probe particles in AdS [1] for large spin states, or a superfluid effective field theory (EFT) for large charge states of certain CFTs [6–9]. The middle of the spectrum also enjoys a natural semiclassical description: thermodynamics [10], and more generally hydrodynamics. The subject of this paper is to study the consequences of this description.

Hydrodynamics is expected to emerge as the late time dynamics of any non-integrable quantum field theory (QFT) at finite temperature. The first theoretically controlled demonstration of this phenomenon is possibly Landau's two-fluid model [11]; for weakly coupled QFTs the emergence of hydrodynamics is now well understood within the framework of Boltzmann kinetic theory [12, 13]. The fluid-gravity correspondence is a more recent example [14–16], for strongly coupled holographic theories. Although an analogous proof in generic CFTs may be too formidable a task for the conformal bootstrap, analytic methods may be able to place constraints on hydrodynamics, such as bounds on transport parameters [17].

The approach followed here is instead to work from the bottom-up, with the hope to guide future efforts from the analytic or numerical bootstrap. Hydrodynamics tightly constrains the thermal correlator of any light neutral operator (e.g. any $\mathbb{Z}_2$-even light operator in the 3d Ising model) at late times. This regime is difficult to address with conventional CFT methods because large Lorentzian times $t \gg \beta$ are far outside of the radius of convergence of the operator product expansion (OPE) [18]. In the microcanonical ensemble, hydrodynamics controls heavy-light four-point functions $\langle HLLH \rangle$ far from the $LL$ OPE limit. Assuming typicality of heavy operators, hydrodynamic predictions can be recast as expressions for off-diagonal heavy-heavy-light OPE coefficients $C_{HH'L}$. Our results, summarized below, should hold in any non-integrable unitary CFT in three or more spacetime dimensions.

## 1.1 Summary of results

We consider thermalizing (or chaotic) CFTs in $d + 1$ spacetime dimensions. Operators that do not carry any internal quantum numbers acquire thermal expectation values: for example a neutral dimension $\Delta_{\mathcal{O}}$ scalar satisfies

$$\langle \mathcal{O} \rangle_\beta = \frac{b_{\mathcal{O}}}{\beta^{\Delta_{\mathcal{O}}}}, \tag{1.1}$$

where $\beta$ is the inverse temperature, and $b_{\mathcal{O}}$ a coefficient that is generically $O(1)$. As argued in Ref. [10], consistency with the microcanonical ensemble implies that diagonal heavy-heavy-light OPE coefficients are on average controlled by the thermal expectation value (1.1). Assuming typicality of heavy eigenstates allows one to drop the averages and leads to the prediction [10] (dropping numerical factors)

$$C_{HH\mathcal{O}} \simeq b_{\mathcal{O}} \left[ \frac{\Delta}{b_T} \right]^{\Delta_{\mathcal{O}}/(d+1)} , \tag{1.2}$$

for the OPE coefficient of two copies of a heavy operator $H$ of dimension $\Delta$ with the light operator $\mathcal{O}$. The dimensionless thermal entropy density $b_T \equiv s\beta^d$ controls the thermal expectation value of the stress-tensor.

In contrast, off-diagonal heavy-heavy-light OPE coefficients $C_{HH'\mathcal{O}}$ should probe out of equilibrium dynamics. If $\mathcal{O}$ is light and the difference in the dimension of the heavy operators is not too large, this will probe the late time, near-equilibrium dynamics, which is controlled by hydrodynamics if $d \geq 2$. Eq. (1.1) shows that $\mathcal{O}$ couples to fluctuations in temperature (or energy density). These propagate as sound, with velocity $c_s^2 = \frac{1}{d}$ and attenuation rate related to the shear viscosity to entropy ratio $\eta_o \equiv \eta/s$ of the CFT, leading to poles near $\omega = \pm k/\sqrt{d}$ in the low frequency $\omega$ and wavevector $k$ thermal two-point function of $\mathcal{O}$

$$\langle \mathcal{O}\mathcal{O} \rangle_{\beta}(\omega, k) \simeq \left( \frac{b_{\mathcal{O}}\Delta_{\mathcal{O}}}{\beta^{\Delta_{\mathcal{O}}}} \right)^2 \frac{\beta^d}{b_T} \frac{\eta_o \beta k^4}{\left( \omega^2 - \frac{1}{d}k^2 \right)^2 + \left( \frac{2d-1}{d}\eta_o \beta \omega k^2 \right)^2} . \tag{1.3}$$

We show under the same assumptions that lead to (1.2) that this hydrodynamic correlator implies

$$|C_{H_J H'_{J'}\mathcal{O}}|^2 \simeq \frac{b_{\mathcal{O}}^2}{e^S} \frac{\eta_o (J-J')^4}{\left[ (\Delta-\Delta')^2 - \frac{1}{d}(J-J')^2 \right]^2 + a_d \, \eta_o^2 \left( \frac{b_T}{\Delta} \right)^{\frac{2}{d+1}} (\Delta-\Delta')^2 (J-J')^4} , \tag{1.4}$$

for the OPE coefficient of the light operator $\mathcal{O}$ with heavy operators of dimension $\Delta$, $\Delta'$ and spin $J$, $J'$. Off-diagonal OPE coefficients are exponentially suppressed in the entropy $S \sim (b_T\Delta^d)^{1/(d+1)}$, as expected on general grounds [4]. We have dropped subexponential dependence on $\Delta$, but instead emphasize the singular dependence on $\Delta-\Delta'$ and $J-J'$ featuring the hydrodynamic sound pole. This result holds for heavy operators satisfying

$$\left( \frac{\Delta}{b_T} \right)^{-\frac{1}{d+1}} \quad \lesssim \quad \Delta - \Delta' \quad \lesssim \quad \left( \frac{\Delta}{b_T} \right)^{\frac{1}{d+1}} . \tag{1.5}$$

The difference in spin must satisfy the same upper bound $J - J' \lesssim (\Delta/b_T)^{1/(d+1)}$. This upper bound comes from the UV cutoff of hydrodynamics, which only describes dynamics at times larger than the thermalization time $t \gtrsim \tau_{\text{th}}$. The lower bound comes from IR effects which resolve the singularity in (1.4). In (1.5) we have assumed $\tau_{\text{th}} \sim \beta$; weakly coupled CFTs have $\tau_{\text{th}} \gg \beta$ and the window (1.5) is parametrically smaller.

Hydrodynamics pervades late time correlators, and not just those of scalar operators. In a thermal state, neutral operators of any integer spin can decay into composite hydrodynamics operators – this is illustrated in Fig. 1. Consider an operator of spin $\ell$. Its component with an even number $\bar{\ell}$ of spatial indices with $2 \leq \bar{\ell} \leq \ell$ has the same quantum numbers as composite hydrodynamic fields involving the stress tensor $T_{\mu\nu}$

$$\mathcal{O}_{i_1 \cdots i_{\bar{\ell}} 0 \cdots 0} \sim \partial_{i_1} \cdots \partial_{i_{\bar{\ell}-1}} T_{0i_{\bar{\ell}}} + T_{0i_1} \partial_{i_2} \cdots \partial_{i_{\bar{\ell}-1}} T_{0i_{\bar{\ell}}} + \cdots . \tag{1.6}$$

This equation is not meant as a microscopic operator equation in the CFT, but rather as an operator equation in the low-energy (dissipative) effective theory around the thermal state. The

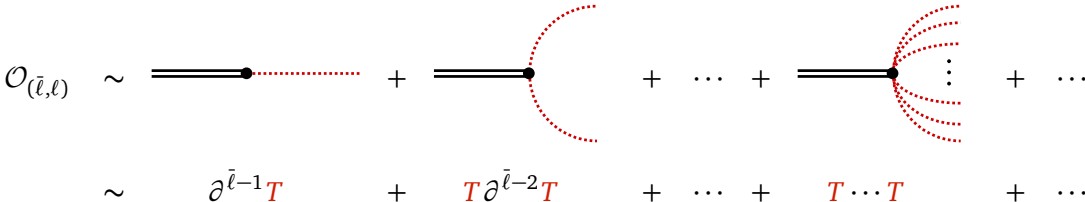

Figure 1: Neutral operators in finite temperature QFT are long-lived as they can decay into hydrodynamic excitations carried by the stress-tensor $T_{\mu\nu}$. $\mathcal{O}_{(\bar{\ell},\ell)}$ denotes components of a spin-$\ell$ operator with $\bar{\ell}$ spatial indices.

first term shows that the operator overlaps linearly with hydrodynamic excitations. Its two-point function will therefore contain hydrodynamic poles, leading to OPE coefficients similar to (1.4). If we consider this operator at vanishing wavevector $k = 0$, then the leading term drops because it is a total derivative and the operator no longer overlaps linearly with hydrodynamic modes. However, it can still decay into the second composite operator which leads to a hydrodynamic loop contribution to its correlator

$$\langle \mathcal{O}_{i_1\cdots i_{\bar{\ell}}0\cdots 0}\mathcal{O}_{j_1\cdots j_{\bar{\ell}}0\cdots 0}\rangle_\beta (t,k=0) \sim \frac{1}{t^{\frac{d}{2}+\bar{\ell}-2}}. \tag{1.7}$$

Although this universal late-behavior for thermal correlators of generic operators in QFTs can be straightforwardly derived using the time-honored framework of fluctuating hydrodynamics, it has to our knowledge not appeared previously in the literature.

The hydrodynamic two-point function (1.7) controls certain OPE coefficients of spinning light operators with two heavy ones, for example when $J = J'$ one finds

$$|C^{\bar{\ell}}_{H_J H'_J \mathcal{O}_\ell}|^2 \simeq e^{-S}\left(\Delta - \Delta'\right)^{\frac{d}{2}+\bar{\ell}-1}, \tag{1.8}$$

for $\bar{\ell}$ even satisfying $2 \leq \bar{\ell} \leq \ell$. Similar results hold for general $\ell$, $\bar{\ell}$, with different exponents in (1.7) and (1.8). The superscript $\bar{\ell}$ on the left-hand side (partially) labels the tensor structure of the spinning OPE. For general spins $J \neq J'$ and $\ell \geq 0$, leading OPE coefficients can be controlled by hydrodynamic correlators at tree-level as in Eq. (1.4), at one-loop as in (1.8), or at higher loop, see Eq. (3.27) for the general expression.

Strictly speaking, the results (1.2), (1.4) and (1.8) hold after averaging the heavy operators over a microcanonical window. However, the expected typicality of heavy operators in generic CFTs imply that a much more sparing averaging may suffice. The eigenstate thermalization hypothesis [19–21] suggests that the diagonal OPE (1.2) holds at the level of individual operators [10], and that the off-diagonal OPEs in e.g. (1.4) and (1.8) hold after averaging over $n$ operators, if one tolerates an error $\sim 1/\sqrt{n}$.

We further derive generalizations of Eqs. (1.4) and (1.8); these results apply to any non-integrable CFT in spatial dimensions $d \geq 2$, without additional continuous global symmetries. Continuous global symmetries $G$ can be incorporated straightforwardly: they lead to additional hydrodynamic modes which can give further contributions to OPE coefficients. We illustrate this with the case $G = U(1)$. OPE coefficients involving charged heavy operators are similar to (1.4) and (1.8), with some differences for odd-spin light operators which receive larger hydrodynamic contributions because of the new slow density. The $U(1)$ symmetry can

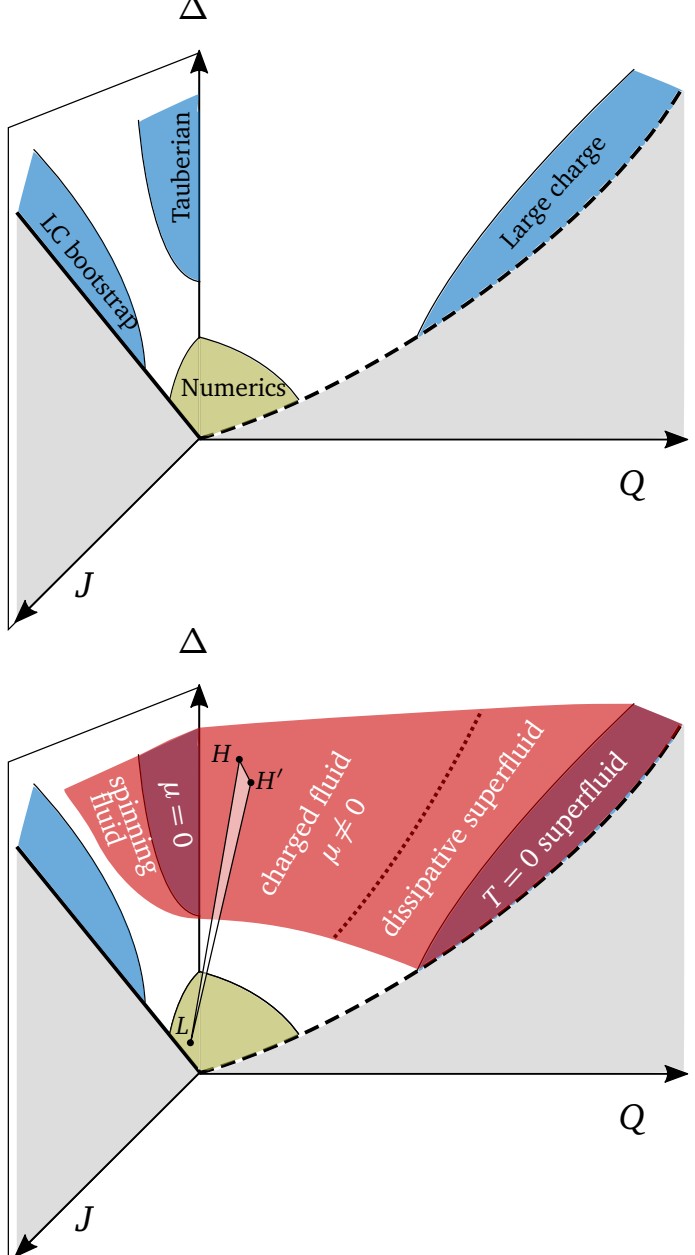

Figure 2: Top: The spectrum of a CFT can be organized using quantum numbers associated with dimension $\Delta$, spin $J$, and internal charge $Q$ if the CFT has additional global symmetries. Existing analytic methods to study various regions of the spectrum include the light-cone bootstrap [1,2], Tauberian theorems [4,5], and the large charge limit [6]. Bottom: The regions that admit a hydrodynamic description are in red. The triangle shows an OPE coefficient $C_{HH'L}$ controlled by hydrodynamics.

be spontaneously broken in the state created by the heavy operator of large charge. In this case, the hydrodynamic description includes a Goldstone phase. This allows us to connect to the large charge program [3, 6–9, 22], which can be thought of as a special case where a hydrodynamic (or semiclassical) description survives the $T \to 0$ limit thanks to the spontaneous breaking of the $U(1)$ symmetry. The various possible phases created by heavy operators are shown in Fig. 2.

The rest of this paper is organized as follows: Fluctuating hydrodynamics is reviewed in Sec. 2, and applied to relativistic QFTs. A few novel results are also obtained there, including the hydrodynamic long-time tails in Eq. (1.7) and a curious aspect of correlation functions $G(t,k)$: these are expected to decay as $e^{-Dk^2 t}$ after the thermalization time in diffusive systems with diffusion constant $D$. However we find that at later times $t \gtrsim \frac{1}{Dk^2} \log \frac{1}{k}$, irrelevant interactions lead to a 'diffuson cascade' with stretched exponential decay $e^{-\sqrt{Dk^2 t}}$. At even later times $t \gtrsim \frac{1}{k^{2d+2}} \log \frac{1}{k}$, perturbative control is lost. In Sec. 3 we study how hydrodynamic correlators control the CFT data, and derive our main results (1.4) and (1.8) along with their generalizations. In Sec. 4, we extend this framework to CFTs with a global $U(1)$ symmetry. We explain how the superfluid EFT can be heated up at small temperatures $1 \ll \beta\mu < \infty$ to connect the hydrodynamic description, and speculate on signatures of thermal phase transitions in the spectrum of heavy operators.

## 2 Hydrodynamics in QFT

Hydrodynamics governs the late time dynamics of non-integrable QFTs at finite temperature. The simplicity of the hydrodynamic description arises from the fact that most excitations are short-lived at finite temperature, with lifetimes of order of the thermalization time $\tau_{\text{th}}$. This allows for an effective description of the system for times

$$t \gg \tau_{\text{th}}, \tag{2.1}$$

in terms long wavelength fluctuations of the variables characterizing thermal equilibrium, namely temperature and velocity $\beta(x)$, $u_\mu(x)$, or their associated densities $T_{00}(x)$, $T_{0i}(x)$. Additional continuous global symmetries would lead to more conserved quantities. These modes are parametrically long lived because their lifetime grows with their wavelength $1/k$. We define the thermalization length $\ell_{\text{th}}$ as the length scale where hydrodynamic modes are no longer parametrically longer-lived than $\tau_{\text{th}}$. We will then focus on modes satisfying

$$k\ell_{\text{th}} \ll 1. \tag{2.2}$$

These time and length scales are parametrically long when the microscopics is weakly coupled, for example $\ell_{\text{th}} \sim \tau_{\text{th}} \sim \frac{\beta}{g^4}$ in (3+1)d gauge theories with coupling $g \ll 1$ [13] . For strongly interacting QFTs (with speed of sound $\sim 1$) one expects $\ell_{\text{th}} \sim \tau_{\text{th}} \gtrsim \beta$, see e.g. [23].

We briefly outline the construction of hydrodynamics for relativistic QFTs, see [24] for a self-contained introduction. Correlation functions for the conserved densities are obtained by solving continuity relations

$$\partial_\mu T^{\mu\nu} = 0. \tag{2.3}$$

These equations also involve the currents $T_{ij}$. They can be closed by writing constitutive relations for the currents in a gradient expansion – in the Landau frame one has

$$\langle T_{\mu\nu}\rangle = \epsilon u_\mu u_\nu + P\Delta_{\mu\nu} - \zeta\Delta_{\mu\nu}\partial_\lambda u^\lambda - \eta\Delta_\mu^\alpha\Delta_\nu^\beta\left(\partial_\alpha u_\beta + \partial_\beta u_\alpha - \frac{2}{d}\eta_{\alpha\beta}\partial_\lambda u^\lambda\right) + \cdots, \tag{2.4}$$

where $P$ is the pressure, $\epsilon$ the energy density, $\zeta$, $\eta$ the bulk and shear viscosities, and the velocity satisfies $u_\mu u^\mu = -1$. We defined the projector $\Delta_{\mu\nu} \equiv \eta_{\mu\nu} + u_\mu u_\nu$. The ellipses denote terms that are higher order in derivatives.

Hydrodynamic correlation functions can be found by expanding fields around equilibrium. These correlation functions are therefore obtained after two expansions: a derivative expansion, apparent in (2.4), and an expansion in fields that we will perform below. The former is always controlled and gives corrections to correlators that are suppressed at late times (2.1), whereas the perturbative expansion in fields is only controlled if interactions are irrelevant – this is the case in $d \geq 2$ spatial dimensions. We first focus on $d > 2$. When $d = 2$, hydrodynamic interactions are only marginally irrelevant [25] – this case will be treated separately in Sec. 2.2. In $d = 1$, interactions are relevant and the theory flows to a new dissipative IR fixed point with dynamic exponent $z = 3/2$ [25–27] (to be contrasted with the unstable diffusive fixed point, where $z = 2$)[†].

When interactions are irrelevant, it is possible to solve Eqs. (2.3) and (2.4) perturbatively in the fields, by expanding around equilibrium

$$u_\mu(x) = \delta_\mu^0 + \delta_\mu^i \frac{\beta}{s} T_{0i} + \cdots, \qquad (2.5a)$$

$$\beta(x) = \beta - \frac{\beta^2 c_s^2}{s} \delta T_{00} + \cdots, \qquad (2.5b)$$

where the entropy density is given by $s = \beta(\epsilon + P)$ and the speed of sound $c_s^2 = \frac{\partial P}{\partial \epsilon}$. This leads to the retarded Green's function

$$G_{T_{00}T_{00}}^R(\omega, k) = \frac{s}{\beta} \left[ \frac{k^2}{c_s^2 k^2 - \omega^2 - i\Gamma_s k^2 \omega} \right] + \cdots \qquad (2.6a)$$

$$G_{T_{0i}T_{0j}}^R(\omega, k) = \frac{s}{\beta} \left[ \frac{k_i k_j}{k^2} \frac{\omega^2}{c_s^2 k^2 - \omega^2 - i\Gamma_s k^2 \omega} + \left( \delta_{ij} - \frac{k_i k_j}{k^2} \right) \frac{Dk^2}{-i\omega + Dk^2} \right] + \cdots, \qquad (2.6b)$$

where $\cdots$ denotes terms that are analytic or subleading when $\omega\tau_{\text{th}}$, $k\ell_{\text{th}} \ll 1$. The long lived densities $T_{00}$, $T_{0i}$ carry a sound mode with attenuation rate $\Gamma_s = \beta \cdot \left( \zeta + \frac{2(d-1)}{d} \eta \right)/s$, and a diffusive mode with diffusion constant $D = \beta \cdot \eta/s$. Other two-point functions can be obtained from the fluctuation-dissipation theorem: the Wightman Green's function for example is $\langle \mathcal{O}\mathcal{O} \rangle(\omega) = \frac{2}{1-e^{-\beta\omega}} \text{Im } G_{\mathcal{O}\mathcal{O}}^R(\omega) \simeq \frac{2}{\beta\omega} \text{Im } G_{\mathcal{O}\mathcal{O}}^R(\omega)$ (here (2.1) implies that we are working at small frequencies $\beta\omega \ll 1$). Its Fourier transform will be used below:

$$\langle T_{0i} T_{0j} \rangle(t, k) = -\frac{s}{\beta^2} \left[ \frac{k_i k_j}{k^2} \cos(c_s k|t|) e^{-\frac{1}{2}\Gamma_s k^2 |t|} + \left( \delta_{ij} - \frac{k_i k_j}{k^2} \right) e^{-Dk^2|t|} \right] + \cdots. \qquad (2.7)$$

For the present purposes it will be useful to understand the constitutive relation (2.4) as an operator equation. Namely, using (2.5) we can write the traceless spatial part as

$$T_{\langle ij \rangle} = -2D\partial_{\langle i} T_{j\rangle 0} + \frac{\beta}{s} T_{0i} T_{0j} - \text{traces} + \cdots. \qquad (2.8)$$

Traceless symmetric combinations are denoted by $A_{\langle ij \rangle} \equiv A_{(ij)} - \frac{1}{d}\delta_{ij} A_k^k$. The operator on the left is studied in the IR by expanding it in terms of composites of IR operators $T_{00}$, $T_{0i}$ with the same quantum numbers (here the quantum number being matched is spin under spatial rotations $SO(d)$). Correlation functions of both operators will match in the IR. This is routinely done in EFTs, e.g. in chiral perturbation theory where UV operators are represented in the IR

---

[†]Neither fixed point describes CFTs in $d = 1$, where the enhanced symmetries completely fix thermal physics in the thermodynamic limit $R/\beta \gg 1$.

in terms of pion degrees of freedom. A similar strategy was followed in [8] where operators with small global charge were represented in terms of operators in the superfluid effective field theory. Although this distinction of UV and IR operators may seem awkward for components of the stress-tensor, we will see that it becomes a useful concept when studying other operators.

In the case at hand, the linear overlap of $T_{\langle ij\rangle}$ with IR degrees of freedom implies that the two-point function of $T_{\langle ij\rangle}$ will contain the hydrodynamic poles in (2.6) (as can be checked explicitly, see e.g. appendix A in Ref. [23]). At $k = 0$, the linear term vanishes, but $T_{\langle ij\rangle}$ can still decay into a composite of hydrodynamic operators via the second term in (2.8). It was found [28] (see [29] for a more recent relativistic exposition) that this term leads to 'long-time tails' in the two-point function

$$
\begin{aligned}
\langle T_{\langle ij\rangle} T_{\langle kl\rangle}\rangle(t, k=0) &\simeq \left(\frac{\beta}{s}\right)^2 \int \frac{d^d k}{(2\pi)^d} G_{T_{0i}T_{0k}}(t, k) G_{T_{0j}T_{0l}}(t, -k) + (i \leftrightarrow j) - \text{traces} \\
&= \frac{A_{ijkl}}{\beta^2 d(d+2)} \left[\frac{1}{(4\pi\Gamma_s|t|)^{d/2}} + \frac{d^2 - 2}{(8\pi D|t|)^{d/2}}\right] + \cdots,
\end{aligned}
\tag{2.9}
$$

where $A_{ijkl} = \delta_{ik}\delta_{jl} + \delta_{il}\delta_{jk} - \frac{2}{d}\delta_{ij}\delta_{kl}$ and the integral was computed using (2.7), dropping terms that decay exponentially fast in time. In the first step, we assumed the theory was Gaussian in the hydrodynamic variables, in which case the symmetric Green's functions factorize [30]. This is of course not the case; the same term in (2.8) that leads to long-time tails is responsible for hydrodynamic interactions (classically, these non-linearities are responsible for turbulence in the Navier-Stokes equations). The framework of fluctuating hydrodynamics addresses these interactions. Although hydrodynamics has been understood as a field theory since the work of Euler, the formulation of dissipative hydrodynamics as an EFT is somewhat more recent [25,31,32] and was motivated by the observation of long-time tails in numerics [33], which are now understood as hydrodynamic loops as in Eq. (2.9). Recent developments in dissipative EFTs for hydrodynamics include [34–37] (see [38] for a review, and e.g. [39–41] for alternative approaches). These constructions allow for a systematic treatment of interactions to arbitrary order in perturbations. Here, we will be working in dimensions where interactions are irrelevant, and will only be interested in the leading hydrodynamic contribution to correlation functions at late times. In this sense we are justified in approximating the action as Gaussian in evaluating (2.9) and in the following. Systematically accounting for corrections to our results would require knowing the structure of interactions in the effective field theory – this was done for simple diffusion in [42].

## 2.1 Late time correlators from hydrodynamics

How do the thermal correlators of other simple operators behave at late times? The central assumption of thermalization and hydrodynamics is that after short time transients, the only long-lived dynamical degrees of freedom are the densities (2.5). Hence any simple operator will be carried by these densities at late times. For example, any neutral spin-2 operator $\mathcal{O}_{\mu\nu}$ will have a constitutive relation similar to (2.4) – the stress tensor is only distinguished by the coefficients in its constitutive relation which are fixed in terms of thermodynamic and transport parameters. More generally, consider a traceless symmetric tensor $\mathcal{O}_{\mu_1\cdots\mu_\ell}$ with even spin $\ell$ (odd spin is mostly similar and is treated in appendix A.2). Its constitutive relation has the schematic form

$$
\begin{aligned}
\mathcal{O}_{\mu_1\cdots\mu_\ell} &= \lambda_0 u_{\mu_1}\cdots u_{\mu_\ell} + \lambda_1 \beta\, \partial_{\mu_1} u_{\mu_2}\cdots u_{\mu_\ell} + \cdots + \lambda_{\ell-1}\beta^{\ell-1}\partial_{\mu_1}\cdots\partial_{\mu_{\ell-1}} u_{\mu_\ell} \\
&\quad + \lambda_\ell \beta^{\ell-1}\partial_{\mu_1}\cdots\partial_{\mu_\ell}\beta + \text{higher derivative},
\end{aligned}
\tag{2.10}
$$

where all terms should be understood to be symmetrized, with traces removed. For some terms there are several possible choices for how the derivatives are distributed – we will be

more precise below after determining which terms are most important. The strategy is simply to write all possible composite hydrodynamic operators with the right quantum numbers, in a derivative expansion – we therefore do not explicitly include terms like $u_{\mu_1}\left(\partial_{\mu_2}\cdots\partial_{\mu_{\ell-1}}\right)\partial^2 u_{\mu_\ell}$ which are manifestly higher order in derivatives. The powers of $\beta$ are chosen such that all coefficients $\lambda$ (which are still functions of $\beta$) have the same dimension, namely that of $\mathcal{O}$ – for CFTs it will be useful to use scale invariance to define instead the dimensionless numbers

$$\lambda_i \equiv b_i/\beta^{2\Delta_\mathcal{O}}. \tag{2.11}$$

The $\lambda_0$ term in (2.10) was considered in a CFT context in [18] – it is special in that it leads to a non-vanishing equilibrium expectation value $\langle\mathcal{O}\rangle_\beta \neq 0$. However, this term will not always give the leading hydrodynamic contribution to the late time correlators of $\mathcal{O}$, as we show below. In particular this term is forbidden by CPT for odd spin $\ell$, but odd spin operators still have hydrodynamic tails.

Let us first consider the components of $\mathcal{O}$ with zero or one spatial index. Linearizing the constitutive relation (2.10) using Eq. (2.5) shows that these components overlap linearly with hydrodynamic modes: the leading terms are

$$\delta\mathcal{O}_{0\cdots0} = -\partial_\beta\lambda_0\frac{\beta^2 c_s^2}{s}\delta T_{00} + \cdots, \tag{2.12a}$$

$$\delta\mathcal{O}_{i0\cdots0} = \lambda_0\frac{\beta}{s}T_{0i} - \lambda_1\frac{\beta^2 c_s^2}{s}\partial_i T_{00} + \cdots. \tag{2.12b}$$

Using (2.6), one finds correlation functions that involve the hydrodynamic poles

$$\langle\mathcal{O}_{0\cdots0}\mathcal{O}_{0\cdots0}\rangle(\omega,k) = \frac{2\beta^d}{s_o}\frac{\left(\beta\partial_\beta\lambda_0\right)^2\Gamma_s c_s^4 k^4}{(\omega^2 - c_s^2 k^2)^2 + (\Gamma_s\omega k^2)^2} + \cdots, \tag{2.13a}$$

$$\langle\mathcal{O}_{i0\cdots0}\mathcal{O}_{j0\cdots0}\rangle(\omega,k) = \frac{2\beta^d}{s_o}\frac{k_i k_j}{k^2}\frac{\left(\omega\lambda_0 + \lambda_1\beta c_s^2 k^2\right)^2\Gamma_s k^2}{(\omega^2 - c_s^2 k^2)^2 + (\Gamma_s\omega k^2)^2}$$
$$+ \frac{2\beta^d}{s_o}\left(\delta_{ij} - \frac{k_i k_j}{k^2}\right)\frac{(\lambda_0)^2 D k^2}{\omega^2 + (D k^2)^2} + \cdots, \tag{2.13b}$$

where we defined the dimensionless entropy density $s_o \equiv s\beta^d$, and $\cdots$ are corrections that are subleading when $\omega\tau_{\text{th}}, k\ell_{\text{th}} \ll 1$.

Now consider correlators involving $\bar{\ell}$ spatial components of the operator $\mathcal{O}$, with $1 < \bar{\ell} \leq \ell$. The constitutive relation (2.10) can again be turned into an operator equation using (2.5) – the part that is traceless symmetric in spatial indices is

$$\mathcal{O}_{\langle i_1\cdots i_{\bar{\ell}}\rangle 0\cdots0} \sim \frac{\lambda_0\beta^{\bar{\ell}}}{s^{\bar{\ell}}}T_{0i_1}\cdots T_{0i_{\bar{\ell}}} + \cdots + \frac{\lambda_{\bar{\ell}-2}\beta^{\bar{\ell}}}{s^2}T_{0i_1}\left(\partial_{i_2}\cdots\partial_{i_{\bar{\ell}-1}}\right)T_{0i_{\bar{\ell}}}$$
$$+ \frac{\lambda_{\bar{\ell}-1}\beta^{\bar{\ell}}}{s}\partial_{i_1}\cdots\partial_{i_{\bar{\ell}-1}}T_{0i_{\bar{\ell}}} + \frac{\lambda_{\bar{\ell}}\beta^{\bar{\ell}+1}c_s^2}{s}\left(\partial_{i_1}\cdots\partial_{i_{\bar{\ell}}}\right)T_{00} + \cdots, \tag{2.14}$$

where again all terms should be understood to be symmetrized, with traces removed. There are still several possibilities for how the derivatives act, e.g. in the $\lambda_{\bar{\ell}-2}$ term – this will be specified shortly. We focus on the traceless symmetric part of $\mathcal{O}_{i_1\cdots i_{\bar{\ell}}0\cdots0}$, because its traces are related to time components of the operator (e.g. $\delta^{i_1 i_2}\mathcal{O}_{i_1\cdots i_{\bar{\ell}}0\cdots} = \mathcal{O}_{00 i_3\cdots i_{\bar{\ell}}0\cdots}$), which in turn satisfy similar constitutive relations with fewer indices. This operator matching equation is illustrated in Fig. 1.

We could now proceed by studying the contribution of every operator in (2.14) to the correlator $\langle\mathcal{O}\mathcal{O}\rangle$. However a simple scaling argument can be used to determine which term in

(2.14) is the most relevant: note that (2.7) implies that the densities scale as $T_{00} \sim T_{0i} \sim k^{d/2}$. For dimensions $d > 2$, it is therefore more advantageous to use gradients to build spin. The most relevant operator is the total derivative term $\lambda_{\bar{\ell}-1}$. We must also keep the term $\lambda_{\bar{\ell}}$; although it is suppressed when $\omega \sim k$ it can give an enhanced contribution when $\omega \lesssim \beta k^2$, as was shown for $\bar{\ell} = 1$ in (2.12b) and (2.13b). Finally, since both of these terms vanish at $k = 0$, it is also important to keep the most relevant operator that is not a total derivative – when $\bar{\ell}$ is even this is $\lambda_{\bar{\ell}-2}$ in (2.14) (when $\bar{\ell}$ is odd, the $\lambda_{\bar{\ell}-2}$ term is a total derivative – this case is treated below). The terms in the constitutive relation (2.14) that give the leading contribution in the hydrodynamic regime $\omega \tau_{\text{th}}, k\ell_{\text{th}} \lesssim 1$ are therefore $\lambda_{\bar{\ell}-2}, \lambda_{\bar{\ell}-1}$ and $\lambda_{\bar{\ell}}$. Which term dominates depends on how $\omega$ compares to the scales $c_s k$ and $Dk^2 \sim \Gamma_s k^2$; their contributions to the correlator take the form[†]

$$
\begin{aligned}
\langle \mathcal{O}_{(\bar{\ell},\ell)} \mathcal{O}_{(\bar{\ell},\ell)} \rangle(\omega, k) = {} & \frac{\beta^d}{s_o} \frac{(\lambda_{\bar{\ell}-1})^2 Dk^2 (\beta k)^{2\bar{\ell}-2}}{\omega^2 + (Dk^2)^2} + \frac{\beta^d}{s_o} \frac{\left(\omega \lambda_{\bar{\ell}-1} + \lambda_{\bar{\ell}} \beta c_s^2 k^2\right)^2 \Gamma_s k^2 (\beta k)^{2\bar{\ell}-2}}{(\omega^2 - c_s^2 k^2)^2 + (\Gamma_s \omega k^2)^2} \\
& + \frac{\beta^d}{s_o^2} \frac{(\lambda_{\bar{\ell}-2})^2}{\omega} \left[ \left(\frac{\omega \beta^2}{\Gamma_s}\right)^{\frac{d}{2}+\bar{\ell}-2} + \left(\frac{\omega \beta^2}{D}\right)^{\frac{d}{2}+\bar{\ell}-2} \right] \left(1 + O\left(\frac{D^2 k^4}{\omega^2}\right)\right) \quad (2.15) \\
& + \cdots.
\end{aligned}
$$

Here we let $\mathcal{O}_{(\bar{\ell},\ell)} \equiv \mathcal{O}_{\langle i_1 \cdots i_{\bar{\ell}} \rangle 0 \cdots 0}$ denote components of a spin $\ell$ operator with $\bar{\ell}$ spatial indices, omitting the corresponding tensor structures; these are treated more carefully in appendix A, see Eq. (A.9). The first line follows from the linear overlaps with the hydrodynamic modes as in (2.13). The second line dominates for $k \to 0$ and comes from a long-time tail contribution to the two-point function from a hydrodynamic loop, as we now explain. The hydrodynamic loop computation is similar to (2.9), with extra gradients acting on the internal legs. Since $\lambda_{\bar{\ell}-2}$ term in (2.14) scales as $k^{d+\bar{\ell}-2}$, one expects a contribution to the two-point function $G_{\mathcal{O}\mathcal{O}}(t) \sim 1/t^{\frac{d}{2}+\bar{\ell}-2}$ (note that one must scale $\omega \sim k^2$). The numerical prefactor can be found by performing the loop integral (see appendix A.1 for more details and the tensor structure):

$$
\langle \mathcal{O}_{(\bar{\ell},\ell)} \mathcal{O}_{(\bar{\ell},\ell)} \rangle(t, k = 0) = \left(\frac{\lambda_{\bar{\ell}-2}}{s_o}\right)^2 \beta^d \left[ \frac{a_1}{(2\Gamma_s |t|/\beta^2)^{\frac{d}{2}+\bar{\ell}-2}} + \frac{a_2}{(4D|t|/\beta^2)^{\frac{d}{2}+\bar{\ell}-2}} \right] + \cdots, \quad (2.16)
$$

where the numerical coefficients $a_1$ and $a_2$ are given in (A.8), and were dropped in (2.15). We see that operators with $\bar{\ell} \geq 2$ spatial indices universally decay as $1/t^{\frac{d}{2}+\bar{\ell}-2}$ in thermalizing QFTs – although this is a straightforward extension of the well-known stress tensor long-time tails (2.9) to operators with higher spin, this result has to our knowledge not appeared previously in the literature. Fourier transforming this result gives the last line in (2.15), where we have also indicated the subleading corrections $O(\frac{D^2 k^4}{\omega^2})$ for small $k \neq 0$ (they are computed explicitly in a special case in appendix A.3, where the analytic structure is also discussed). When the number of spatial derivatives $\bar{\ell}$ is odd, the $\lambda_{\bar{\ell}-2}$ term in the constitutive relation (2.14) is a total derivative, and there is competition between less relevant terms. Their contribution to the late time correlator can be computed as in the even $\bar{\ell}$ case – for $\bar{\ell} \geq 3$, one finds

$$
\langle \mathcal{O}_{(\bar{\ell},\ell)} \mathcal{O}_{(\bar{\ell},\ell)} \rangle(t, k = 0) \sim \frac{1}{|t|^{\alpha_{\bar{\ell}}}} \quad \text{with} \quad \alpha_{\bar{\ell}} = \begin{cases} d + \bar{\ell} - 3 & \text{if } d \leq 4, \\ \dfrac{d}{2} + \bar{\ell} - 1 & \text{if } d > 4. \end{cases} \quad (2.17)
$$

---

[†]These hydrodynamic contributions also imply that correlators $\langle \mathcal{O}_{(\bar{\ell},\ell)} \mathcal{O}_{(\bar{\ell},\ell)} \rangle(x, t)$ of neutral operators always decay polynomially in real time. Exponential decay of correlators is therefore not a good criterion for thermalization. I thank Erez Berg for discussions on this point.

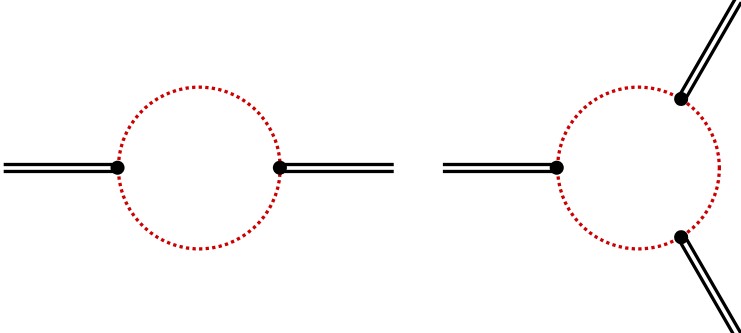

Figure 3: Hydrodynamic loops control the correlators of $k = 0$ neutral operators at large time separation.

See appendix A.2 for more details. The first line in (2.15) is then unchanged for $\bar{\ell}$ odd, but the second line will be given by the Fourier transform of (2.17) instead of (2.16).

In theories with a large number of degrees of freedom such as holographic theories, the suppression in (2.16) by the dimensionless entropy density $s_o \equiv s\beta^d \sim N^2 \gg 1$ implies that these hydrodynamic tails will only overcome short-time transients $\sim e^{-t/\tau_{\text{th}}}$ at times $t \gtrsim \tau_{\text{th}} \log s_o$ (the late time limit of correlation functions therefore does not commute with the $N \to \infty$ limit). The stress tensor tails (2.9) were captured in a holographic model in Ref. [43] by computing a graviton loop in the bulk. However certain tails in the holographic correlators of higher-spin operators (2.16) are reproduced more simply, and are direct consequences of large $N$ factorization: consider a holographic model with a single trace scalar $\phi$. In the absence of a $\phi \to -\phi$ symmetry, the scalar will have a thermal expectation value

$$\langle \phi \rangle_\beta = \frac{b_\phi}{\beta^{\Delta_\phi}} \, . \tag{2.18}$$

This is achieved in bottom-up holographic models by including a coupling in the bulk between the scalar and the Weyl tensor [44] (see also [45, 46]). A computation of the scalar two-point function should reveal the sound mode as in (2.13a). Double-trace spin-$\ell$ operators $\mathcal{O}_\ell \sim \phi \partial^\ell \phi$ will then have long-time tail contributions to their thermal correlators, similar to (2.16).

Although $s_o$ acts as a loop counting parameter in fluctuating hydrodynamics, we emphasize that the perturbative expansion is controlled even when $s_o \sim 1$ because hydrodynamic interactions are irrelevant[†]. In this paper, we do not assume that $s_o$ is large.

We focused above on diagonal two-point functions; extending these results to off-diagonal correlators $\langle \mathcal{O}_{(\bar{\ell},\ell)} \mathcal{O}'_{(\bar{\ell}',\ell')} \rangle$ is straightforward, see appendix A.1. These methods can also be easily extended to compute thermal higher-point correlators, which at large time separations are also controlled by a hydrodynamic loop, see Fig. 3. For example, operators $\mathcal{O}_{(\bar{\ell},\ell)}(t, k = 0)$ with an even number of spatial indices $\bar{\ell}$ have a symmetric connected $n$-point function with $n \geq 3$ odd given by

$$\langle \mathcal{O}_{(\bar{\ell},\ell)}(t_1) \cdots \mathcal{O}_{(\bar{\ell},\ell)}(t_n) \rangle_c \sim \left( \frac{\lambda_{\bar{\ell}-2}}{s_o} \right)^n \frac{\beta^{d(n-1)}}{[D(t_{12} + t_{23} + \cdots + t_{n1})/\beta^2]^{\frac{d}{2} + \frac{n}{2}(\ell-2)}} + \text{sym.} \,, \tag{2.19}$$

with $t_{ij} \equiv |t_i - t_j|$, and where 'sym.' means symmetrizing[‡] the times $t_1, \cdots, t_n$. When $n$ is odd, the contribution from the sound pole vanishes because the integrand $\cos^n(c_s k|t|)$ oscillates

---

[†]For example, the quark-gluon plasma has $s_o \sim 10$ [47–49].

[‡]In the approach presented here, the correlators are necessarily symmetrized. Correlators with arbitrary time orderings can however still be computed from hydrodynamics using the effective action, see [50].

around zero; when $n$ is even the correlator receives an extra contribution from the sound attenuation rate as in (2.16).

## 2.2 The critical dimension $d = 2$

The results in the previous section apply to any QFT in spatial dimensions $d > 2$. For $d = 2$, hydrodynamic interactions are only marginally irrelevant. One manifestation of this is that all terms in the first line of (2.14) have the same scaling. This implies that many terms contribute to the correlator, which however still scales like (2.16).

An additional subtlety is that the transport parameters $D$, $\Gamma_s$ now run. For simplicity, let us assume the bulk viscosity $\zeta = 0$ (as is the case for CFTs), so that $\Gamma_s = D = \beta \eta/s$. The $\beta$-function for $D$ is negative [24, 25], so that it flows to infinity in the IR[†]. Indeed, the tree-level and one-loop contribution to the Green's function can be found from (2.6) and (2.9) to be

$$G_{T_{xy} T_{xy}}(\omega, k = 0) = \frac{2s}{\beta^2} \left[ D + \frac{1}{16\pi s D} \log \frac{1}{\omega} + \cdots \right].$$
(2.20)

Interpreting the quantity in brackets $D + \frac{\partial D}{\partial \log \omega} \log \omega$ as a running of the diffusion constant one finds [25]

$$D(\omega) \simeq D_\Lambda \sqrt{1 + \frac{\log \Lambda/\omega}{8\pi s D_\Lambda^2}},$$
(2.21)

where $D_\Lambda$ is the diffusion constant at the scale $\Lambda$. In the deep IR

$$D(\omega) \simeq \sqrt{\frac{\log 1/\omega}{8\pi s}}.$$
(2.22)

It is a striking feature of (2+1)d hydrodynamics that dissipation does not introduce new parameters at the latest times – transport parameters are fixed in terms of the thermodynamics [24][‡]. In practice, the asymptotic value may only be reached at very late times, or small frequencies. Taking $\Lambda = 1/\beta$ and assuming $D_\Lambda \approx \beta$ one needs frequencies $\beta\omega \lesssim e^{-8\pi s_o}$ for the asymptotic diffusion (2.22) to be reached, where the dimensionless entropy density[§] $s_o \equiv s\beta^2$. These logarithmic corrections to transport propagate to correlation functions of generic operators, so that transport parameters in e.g. Eq. (2.15) will be replaced with (2.21) – however since many other terms in (2.14) contribute to the same order in $\omega$ and $k$ when $d = 2$, we will not attempt to obtain the exact correlator. These logarithmic corrections are negligible for many practical purposes, but have been observed in classical simulations, see e.g. [53, 54]. We will mostly ignore logarithmic corrections in applications to CFT data in Sec. 3.

## 2.3 Real time correlators and diffuson cascade

The hydrodynamic correlators in frequency space $G(\omega, k)$ obtained above are the ingredients needed for the CFT applications in Sec. 3; the reader interested in these results may therefore directly skip ahead to that section. In this section we take a slight digression to discuss finite temperature QFT correlators in real time. At finite wavevector $k$, the linearized hydrodynamic correlators (2.7) decay exponentially in time $\sim e^{-Dk^2|t|}$. We will see in this section that even this standard result is drastically affected by hydrodynamic fluctuations, which in a sense are

---

[†]This implies that canonically normalized interactions $\sim 1/D$ are marginally irrelevant and the theory is 'free' in the IR, in the sense that it is described by regular tree level hydrodynamics like in higher dimensions.

[‡]Dissipation is also tied to thermodynamics in (1+1)d when hydrodynamic fluctuations are relevant [25–27], as was recently emphasized in Ref. [51].

[§]For CFTs, $s_o = b_T$ in the notation of [18]. A free massless scalar has $s_o = \frac{3\zeta(3)}{2\pi}$, so that $e^{-8\pi s_o} \approx 5 \times 10^{-7}$. For the (2 + 1)d Ising model $s_o \approx 0.459$ [18, 52] so $e^{-8\pi s_o} \approx 10^{-5}$.

dangerously irrelevant: although they only give small corrections to $G(\omega, k)$, they entirely control the leading behavior of $G(t, k)$ at late times.

Let us therefore study the real time thermal correlation function of an operator with $\bar{\ell}$ spatial indices

$$G(t, k) \equiv \langle \mathcal{O}_{(\bar{\ell}, \ell)} \mathcal{O}_{(\bar{\ell}, \ell)} \rangle(t, k). \tag{2.23}$$

We will take $\bar{\ell} \geq 2$ even for simplicity, but similar results hold for any $\bar{\ell}$. Based on the previous section, one expects the polynomial decay of correlation functions (2.16) to still hold for times smaller than the diffusion time $1/Dk^2$ of the mode

$$1 \quad \ll \quad \frac{t}{\tau_{\text{th}}} \quad \ll \quad \frac{1}{Dk^2\tau_{\text{th}}} \quad \sim \quad \frac{1}{(k\ell_{\text{th}})^2}, \tag{2.24}$$

(in this section, we assume for simplicity that $D \sim \Gamma_s \sim \ell_{\text{th}}^2/\tau_{\text{th}}$). The small wavevector of the operator in units of the UV cutoff of hydrodynamics $k\ell_{\text{th}} \ll 1$ will allow for a parametric separation between various regimes of the correlator at late times. To find the cross-over time more precisely, we can compare the contributions from the two first terms in Fig. 1 to the two-point function – one finds that the polynomial decay (2.16) holds in the window

$$\text{regime I:} \qquad G(t, k) \sim \frac{1}{t^{\frac{d}{2}+\bar{\ell}-2}}, \qquad 1 \quad \ll \quad \frac{t}{\tau_{\text{th}}} \quad \ll \quad \frac{1}{(k\ell_{\text{th}})^{2-\gamma}}, \tag{2.25}$$

with $\gamma = 2\frac{d-2}{d+2\bar{\ell}-4} \in (0, 2)$. At slightly later times, the correlator is controlled by the linear overlap with the hydrodynamic mode and has the form

$$\text{regime II:} \quad G(t, k) \sim k^{2\bar{\ell}-2}e^{-Dk^2|t|}, \quad \frac{1}{(k\ell_{\text{th}})^{2-\gamma}} \ll \frac{t}{\tau_{\text{th}}} \ll \frac{1}{(k\ell_{\text{th}})^2}\log\frac{1}{(k\ell_{\text{th}})^d}. \tag{2.26}$$

The power-law decay therefore plateaus to a constant before starting to decay exponentially around the diffusion time $1/Dk^2$. So far the discussion here mirrors the one in frequency space, see Eq. (2.15). However the result above eventually breaks down at late times. Indeed, consider again the second term in Fig. 1, where the operator decays into two hydrodynamic excitations. Since it is less relevant than the first term, its contribution to the correlator will be more suppressed by $1/t$ (or $k$); however its exponential factor is larger:

$$G_{T\partial^{\bar{\ell}-2}T, T\partial^{\bar{\ell}-2}T}(t, k) \sim k^{2\bar{\ell}-2}k^{d-2}e^{-\frac{1}{2}Dk^2|t|}. \tag{2.27}$$

The exponent corresponds to the energy threshold for production of two diffusive fluctuations, which is *half* that of a single diffusive mode [42]. More generally, the operator $\mathcal{O}(k, t)$ can decay into $n$ diffusive modes, distributing its momentum such that each mode carries $k' = k/n$ so that the exponential factor becomes $\left(e^{-Dk'^2t}\right)^n = e^{-\frac{1}{n}Dk^2t}$. This is the manifestation in real time $t$ of the $n$-diffuson branch cut, with branch point $\omega_{n\text{-diff}} = -\frac{i}{n}Dk^2$. There are similar branch points at the threshold for production of $n$ sound modes $\omega_{n\text{-sound}} = \pm c_s k - \frac{i}{2n}\Gamma_s k^2$ (see appendix A.3). The analytic structure of $G(\omega, k)$ is shown in Fig. 4.

For $n$ sufficiently large, the $n$-diffuson contribution to the correlator has the form[†]

$$G(t, k)|_{n\text{-diff}} \sim n!(k\ell_{\text{th}})^{nd}e^{-\frac{1}{n}Dk^2t}. \tag{2.28}$$

The perturbative expansion in $k\ell_{\text{th}}$ is presumably asymptotic and this result should therefore only be trusted for $n \lesssim 1/(k\ell_{\text{th}})^d$. The largest contribution can be determined by extremizing

---

[†]This expression applies when $n$ is larger than the spin $\ell$. When $n \lesssim \ell$, decaying into one more $T_{0i}$ costs $k^{d/2}$ but saves a derivative $k$, so that the suppression is only $(k\ell_{\text{th}})^{n(d-2)}$ instead of $(k\ell_{\text{th}})^{nd}$ in (2.28).

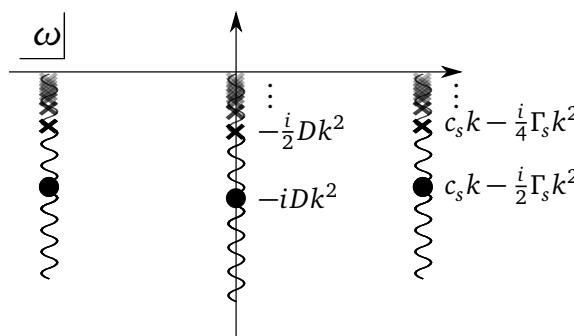

Figure 4: Analytic structure of hydrodynamic correlation functions $G^R(\omega, k)$. The circles denote hydrodynamics poles $\omega_{\text{diff}} = -iDk^2$ and $\omega_{\text{sound}} = \pm c_s k - \frac{i}{2}\Gamma_s k^2$, and the crosses denote branch points $\omega_{n\text{-diff}} = -\frac{i}{n}Dk^2$ and $\omega_{n\text{-sound}} = \pm c_s k - \frac{i}{2n}\Gamma_s k^2$ located at the threshold for production of $n$ hydrodynamic excitations.

(2.28) over $n$. Approximating $\log n! \sim n \log n$ and using $n \ll 1/(k\ell_{\text{th}})^d$ one finds that the largest contribution at time $t$ comes from decay into

$$n(t) \simeq \sqrt{\frac{Dk^2 t}{d \log \frac{1}{k\ell_{\text{th}}}}} \tag{2.29}$$

diffusons. Plugging back into (2.28) produces the correlator

$$\text{regime III:} \quad G(t, k) \sim e^{-\alpha\sqrt{Dk^2|t|}}, \quad \frac{1}{(k\ell_{\text{th}})^2}\log\frac{1}{k\ell_{\text{th}}} \ll \frac{t}{\tau_{\text{th}}} \ll \frac{1}{(k\ell_{\text{th}})^{2d+2}}\log\frac{1}{k\ell_{\text{th}}}, \tag{2.30}$$

with $\alpha \sim \sqrt{d \log \frac{1}{k\ell_{\text{th}}}} \sim 1$. In this regime, operators decay into more and more diffusive excitations, which leads to a stretched exponential decay of correlators.

Although we have focused on the decay of operators with $\bar{\ell}$ spatial indices in QFTs at finite temperature, similar results apply for two-point functions of generic neutral operators in any diffusive system, including non-integrable spin chains and random unitary circuits with conservation laws. In particular Eqs. (2.26) and (2.30) would apply there, after removing the spatial spin dependence $k^{2\bar{\ell}-2} \to 1$. Certain signatures of diffusive tails have been observed numerically in these systems [55, 56], and finite $k$ correlators have been studied e.g. in [57], but to our knowledge this diffuson cascade (2.30) and cross-over from $e^{-Dk^2 t}$ to $e^{-\sqrt{Dk^2 t}}$ has yet to be observed. One issue is that of finite system size, which we discuss below.

In the thermodynamic limit, correlators will decay as (2.30) as long as the perturbative expansion of fluctuating hydrodynamics holds. Given that Eq. (2.28) explodes for decay into $n \gg 1/(k\ell_{\text{th}})^d$ diffusons, we expect the hydrodynamic expansion for $G(t, k)$ to breakdown at times $t \gtrsim t_{\text{breakdown}}$, with

$$n(t_{\text{breakdown}}) \sim \frac{1}{(k\ell_{\text{th}})^d} \qquad \Rightarrow \qquad \frac{t_{\text{breakdown}}}{\tau_{\text{th}}} \sim \frac{1}{(k\ell_{\text{th}})^{2d+2}}\log\frac{1}{k\ell_{\text{th}}}, \tag{2.31}$$

which is therefore the upper limit of regime III in (2.30). We do not know of a controlled way to compute hydrodynamic correlation functions $G(t, k)$ at times $t \gtrsim t_{\text{breakdown}}$.

In a finite volume $L^d$ there is a minimal wavelength that the diffusive fluctuations in the loop can carry: $k_{\text{min}} = \frac{2\pi}{L}$. The correlator will then be controlled by decay of the operator into $n_{\text{max}} \sim k/k_{\text{min}}$ diffusive modes with momentum $k_{\text{min}}$, so that at times later than the Thouless time $L^2/D$ the correlation function has the form

$$\text{regime IV:} \quad G(t, k) \sim e^{-D|k||k_{\text{min}}||t|}, \quad \frac{L^2}{\ell_{\text{th}}^2} \ll \frac{t}{\tau_{\text{th}}} \ll \frac{sL^{d+1}}{k\ell_{\text{th}}^2}, \tag{2.32}$$

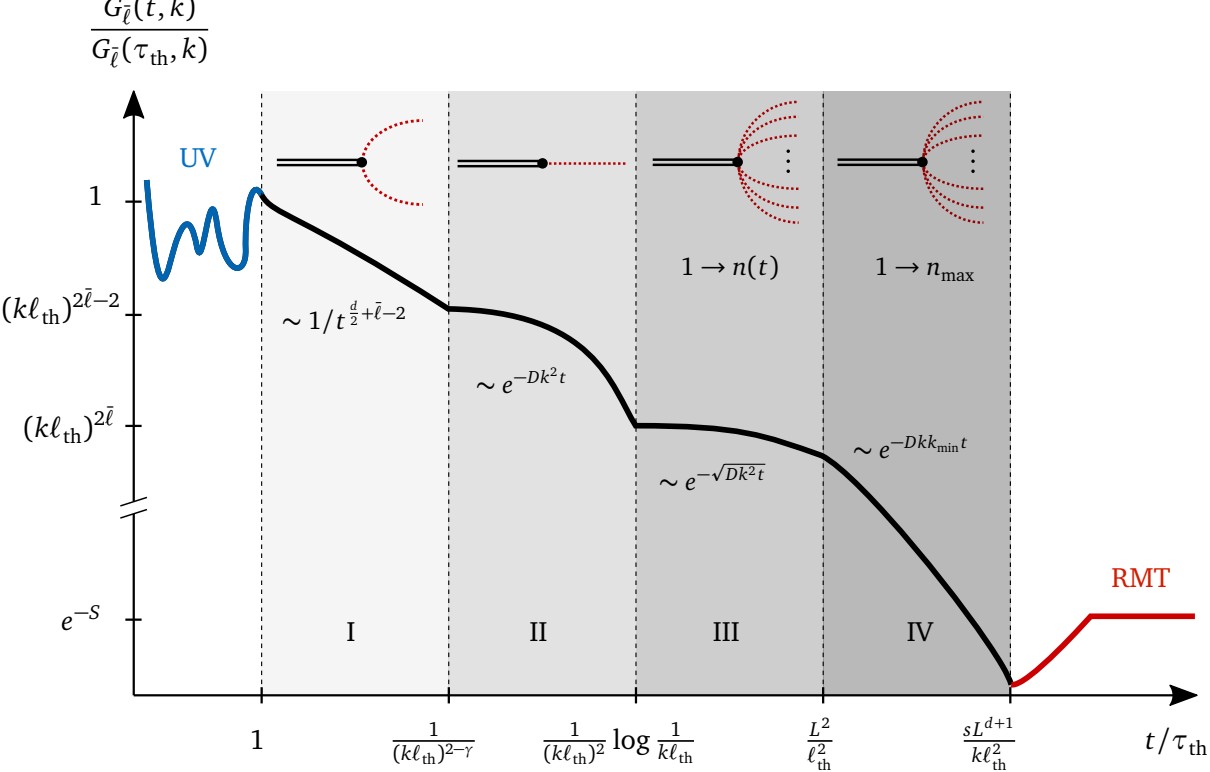

Figure 5: Schematic log-log plot of late time two-point functions (2.23) in interacting QFTs at finite temperature. The polynomial decay in regime I depends on the spatial spin of the operator, however regimes II-IV should occur in any diffusive system without the notion of spatial spin, such as non-integrable spin chains. The small wavevector of the operator $k\ell_{\text{th}} \ll 1$ allows for a parametric separation of the four hydrodynamic regimes. We have assumed that the Thouless time occurs before the breakdown time (2.31).

where $s$ is the entropy density. Here we are assuming that the Thouless time occurs before the breakdown of hydrodynamics (2.31). When $t$ reaches the upper limit, the Green's function is exponentially small $\sim e^{-S}$. At this point we expect the correlator to be described by random matrix theory (RMT) [58,59], exhibiting a ramp that levels off to a plateau[†], upon averaging (over a few operators with the same quantum numbers, for example). The exponentially small value of the Green's function

$$G \sim e^{-S} \sim \exp\left[-\frac{s_o}{(k_{\min}\beta)^d}\right],\qquad(2.33)$$

shows that RMT effects are non-perturbative in the hydrodynamics description, which is an expansion in $k\ell_{\text{th}} \sim k\beta$ (2.2). Fig. 5 summarizes the various regimes of the correlator. To our knowledge, regimes I, III and IV have not appeared previously in the literature.

We emphasize that these results hold for any non-integrable QFT, with the regimes II, III and IV holding more generally for any diffusive system. The microscopic couplings only enter in the determination of the thermalization time $\tau_{\text{th}}$, and transport parameters such as $D$. For weakly coupled theories, the early time behavior $t \ll \tau_{\text{th}}$ can be studied using direct finite temperature perturbation theory or kinetic theory [13] (which can also capture chaos

---

[†]Note that the onset time of a RMT description $t_{\text{RMT}} \sim \frac{sL^{d+1}}{k\ell_{\text{th}}^2}$ depends on the observable, here through its wavevector $k$. Onset of RMT in the spectral form factor is expected to happen at earlier times: in $d = 1$, $t_{\text{RMT}}^{\text{SFF}} \sim \frac{L^2}{D}$ [59] is smaller than the time scale above by a factor $k/s \lesssim k\ell_{\text{th}} \ll 1$.

[60]). However it is difficult to observe the regimes I, III and IV directly in a weakly coupled approach, as hydrodynamic fluctuations are not captured by the linearized approximation to the Boltzmann kinetic equation.

We close with a comment on the convergence of the perturbative expansion. The convergence of the hydrodynamic gradient expansion in large $N$ systems (where hydrodynamic interactions can be ignored if one takes the $N \to \infty$ limit first) has been discussed e.g. in [61,62]. Away from the large $N$ limit, one expects that loop effects cause the gradient expansion to be asymptotic, as usual in effective field theories. This is apparent in the $n$-diffuson contribution to the correlator (2.28), which blows up when $n \gg 1/(k\ell_{\text{th}})^d$. It would be interesting to understand if this explosion can be tamed, or Borel resummed, to produce a prediction for correlators $G(t,k)$ after the breakdown time (2.31). In perturbative QFT, processes involving many particles also lead to a breakdown of the perturbative expansion, which can however be saved by expanding around a different saddle [63,64] (see [65] for recent developments); diffusive systems are a natural venue to study multiparticle processes, and perhaps apply some of these techniques.

# 3 Semiclassical theory of heavy operators in CFTs

We found in the previous section that the late time thermal two-point functions of light neutral operators of any spin are governed by hydrodynamics in generic (thermalizing) CFTs. Working in the microcanonical ensemble, this implies that off-diagonal heavy-heavy-light OPE coefficients $C_{HH'L}$ are universal, at least on average. A priori, the averaging must be done over a microcanonical window of states. However, heavy operators in thermalizing CFTs are expected to look typical, so that much less averaging may be needed in practice. This expectation is objectified by the ETH Ansatz [10, 19–21] for the matrix elements of a light local operator $\mathcal{O}$ in energy-momentum eigenstates[†] $\hat{P}_\mu |H\rangle = |H\rangle p_\mu$ :

$$\langle H'|\mathcal{O}|H\rangle = \langle \mathcal{O}\rangle_\beta \delta_{HH'} + \Omega(p)^{-1/2} R^{\mathcal{O}}_{HH'} \sqrt{\langle \mathcal{O}\mathcal{O}\rangle (p-p')}, \tag{3.1}$$

where $\Omega(p)$ is the density of states at momentum $p$, and the $R^{\mathcal{O}}_{HH'}$ behave like independent random variables with unit variance. Averaging Eq. (3.1) over a microcanonical window of heavy operators $H, H'$ simply states the equivalence between microcanonical and canonical ensembles; the non-trivial content of Eq. (3.1) is instead that microcanonical averaging is unnecessary: diagonal matrix elements directly produce thermal expectation values, and off-diagonal matrix elements probe out of equilibrium response, for example through symmetric two-point function $\langle \mathcal{O}\mathcal{O}\rangle$. The appearance of $\langle \mathcal{O}\mathcal{O}\rangle$ in the variance above is required for the Ansatz to reproduce the two-point function [21, 23] (note that the Wightman and symmetric two-point functions are approximately equal in the hydrodynamic regime (2.1)). In a CFT, the state-operator correspondence relates these matrix elements to OPE coefficients. For scalar operators (see e.g. [4])

$$C_{HH'\mathcal{O}} = R^{\Delta_{\mathcal{O}}} \langle H|\mathcal{O}|H'\rangle. \tag{3.2}$$

The diagonal part of ETH (3.1) implies that diagonal heavy-heavy-light OPEs are controlled by equilibrium thermodynamics, as found in Ref. [10] (see also [4,66]). In section 3.1 their results are reviewed and extended to operators with spin. In section 3.2 we turn to the off-diagonal part of (3.1), and show how hydrodynamics controls the corresponding OPE coefficients.

---

[†]More precisely, the energy of the heavy state on the cylinder $\mathbb{R} \times S^d$ is $p_0$ and $p_i$ labels the spherical harmonic on the spatial sphere. We will mostly focus on regimes where the sphere can be approximated as $S^d \to \mathbb{R}^d$ (see Eq. (3.11) and comment below), so that $p_i = k_i$ will denote regular spatial momentum.

## 3.1 Thermodynamics in OPE data

Consider a heavy operator $H$, with dimension $\Delta \equiv \Delta_H \gg 1$ larger than any other intrinsic number of the CFT (such as measures of the number of degrees of freedom). It will be useful to define the energy density $\epsilon$ of the state that $H$ creates on the cylinder $\mathbb{R} \times S^d$ of radius $R$

$$\Delta = \epsilon R^{d+1} S_d \,, \tag{3.3}$$

where $S_d \equiv \text{Vol} S^d = \frac{2\pi^{(d+1)/2}}{\Gamma(\frac{d+1}{2})}$. We can then reach the macroscopic limit by taking $R \to \infty$ while keeping $\epsilon$ fixed. The diagonal OPE coefficient is fixed by thermodynamics [10]

$$C_{HH\mathcal{O}} = R^{\Delta_\mathcal{O}} \langle H | \mathcal{O} | H \rangle \simeq b_\mathcal{O} (R/\beta)^{\Delta_\mathcal{O}} = b_\mathcal{O} \left[ \frac{d+1}{dS_d} \frac{\Delta}{b_T} \right]^{\Delta_\mathcal{O}/(d+1)} \,, \tag{3.4}$$

where in the second step we used (3.1), and (3.3) in the last to eliminate the radius. We used (2.10) and (2.11) to express thermal expectation values as

$$\langle \mathcal{O} \rangle_\beta = \frac{b_\mathcal{O}}{\beta^{\Delta_\mathcal{O}}} \,, \qquad \langle T_{\mu\nu} \rangle_\beta = \frac{b_T}{\beta^{d+1}} \left( \delta_\mu^0 \delta_\nu^0 + \frac{\eta_{\mu\nu}}{d+1} \right) \,, \tag{3.5}$$

where $b_T = s_o = s\beta^d$ is the dimensionless entropy density, and is related to the energy density as $\epsilon = \frac{d}{d+1} b_T / \beta^{d+1}$.

Eq. (3.4) can be straightforwardly extended to operators with spin. From Eq. (2.14) we find that the thermal expectation value of light operator of even spin $\ell$ takes the form

$$\langle \mathcal{O}_{\mu_1 \cdots \mu_\ell} \rangle_\beta = \frac{b_\mathcal{O}}{\beta^{\Delta_\mathcal{O}}} \left( \delta_{\mu_1}^0 \cdots \delta_{\mu_\ell}^0 - \text{traces} \right) \,, \tag{3.6}$$

where we again used scale invariance to write $\lambda_0 = b_\mathcal{O}/\beta^{\Delta_\mathcal{O}}$. If the heavy operators are still scalars, the OPE coefficient $C_{HH\mathcal{O}_\ell}$ is parametrized by a single tensor structure [67], which agrees with (3.6) (see Ref. [3] for similar checks in the large charge limit). Now if the heavy operators carry a spin $J$ that is not macroscopic – i.e. $\frac{J}{\Delta} \to 0$ in the macroscopic limit – the states they create on the cylinder are homogeneous so that (3.6) still applies. However, many tensor structures can now appear [67], each with their own OPE coefficients. For example, the OPE coefficient involving heavy states $|H, Jm\rangle$ in an irreducible representation $J$ of the Lorentz group with weight $|m| \le J$

$$\langle H, Jm | \mathcal{O}_\ell | H, Jm \rangle \sim C_{H_J H_J \mathcal{O}_\ell}^m \tag{3.7}$$

could depend on $J$ and $m$ (we are using $SO(3)$ notation for simplicity, which can be easily generalized to $SO(d+1)$ for $d > 2$). Note that we are focusing on diagonal matrix elements here, so that both states have to have the same weight. Comparison with (3.6) shows that the leading answer is in fact independent of $J$ and $m$ as long as $J \ll \Delta$, so that (3.4) still holds

$$C_{H_J H_J \mathcal{O}_\ell}^m \simeq b_\mathcal{O} \left[ \frac{d+1}{dS_d} \frac{\Delta}{b_T} \right]^{\Delta_\mathcal{O}/(d+1)} \,. \tag{3.8}$$

Diagonal OPE coefficients involving heavy operators with macroscopic spin $J \sim \Delta$ are discussed in section 3.3.

## 3.2 Hydrodynamics in OPE data

We will use (3.1) and the correlators obtained in Sec. 2 to determine OPE coefficients (3.2) in the 'macroscopic' limit $R \to \infty$, with a 'mesoscopic' difference in the dimensions of the heavy operators $\Delta \equiv \Delta_H$, $\Delta' \equiv \Delta_{H'}$, namely

$$\Delta \simeq \Delta' = \epsilon R^{d+1} S_d \,, \qquad \Delta - \Delta' = \omega R \,, \tag{3.9}$$

in spacetime dimensions $d+1 \geq 3$, keeping the energy density $\epsilon$ and frequency $\omega$ finite. When the mesoscopic difference in their dimensions is not large

$$\omega = \frac{\Delta - \Delta'}{R} \ll \frac{1}{\tau_{\text{th}}} \sim \frac{1}{\beta}, \tag{3.10}$$

the off-diagonal OPE coefficient $C_{HH'\mathcal{O}}$ is controlled by hydrodynamics. In the last step we have assumed the CFT is strongly coupled, so that the thermalization time is set by the temperature – in a weakly coupled CFT the frequency window where hydrodynamic applies is parametrically suppressed,[†] see discussion below (2.2). Eliminating the radius, this hydrodynamic window is

$$\left(\frac{\Delta}{b_T}\right)^{-\frac{1}{(d+1)}} \lesssim \Delta - \Delta' \lesssim \left(\frac{\Delta}{b_T}\right)^{\frac{1}{(d+1)}}. \tag{3.11}$$

The lower bound comes from the fact that the hydrodynamic results will receive corrections from the finite size of the sphere of radius $R$ at the Thouless energy $\omega \sim D/R^2$ – these could be obtained by generalizing the hydrodynamic correlators of Sec. 2 to the sphere, but we will not attempt to do so here; we expect the singular features that we find in OPE coefficients to be softened in that regime. The upper bound however is a fundamental UV cutoff of hydrodynamics (2.1), assuming $\tau_{\text{th}} \sim \beta$.

The OPE coefficients $C_{H_J H'_{J'} \mathcal{O}_\ell}$ will depend on the quantum numbers of the heavy operators $\Delta, \Delta', J, J'$, those of the light operator $\Delta_{\mathcal{O}}, \ell$, the thermal properties of the CFT through $b_T$ and $\eta_o \equiv \eta/s$, and finally on the thermal properties of the light operator through the coefficients $b_i$ in (2.11). In CFTs, the hydrodynamic correlators of Sec. 2 simplify somewhat, because tracelessness of the stress tensor forces the bulk viscosity to vanish $\zeta = 0$ and fixes the speed of sound $c_s^2 \equiv \frac{\partial P}{\partial \epsilon} = \frac{1}{d}$. The diffusion constant and sound attenuation rate in (2.6) or (2.15) are therefore given by

$$D = \eta_o \beta, \qquad \Gamma_s = \frac{2(d-1)}{d} \eta_o \beta. \tag{3.12}$$

The simplest case of three scalar operators $J = J' = \ell = 0$ is somewhat subtle and will be discussed further below. Instead we start with a light spinning operator with $\ell \geq 2$, and take $J, J'$ 'microscopic', i.e. they are kept fix $\sim 1$ in the macroscopic limit $R \to \infty$.

### 3.2.1 Microscopic spin $J, J'$

Keeping the spin $J, J' \sim 1$ fixed in the $R \to \infty$ limit implies that the Green's function in (3.1) must be evaluated at spatial wave-vector $k = \frac{J-J'}{R} = 0$. In this case we found that the correlator is controlled by hydrodynamic loops: for components with $\bar{\ell} \geq 2$ spatial indices, the Green's function is (using (2.15) and (2.11))

$$\langle \mathcal{O}_{(\bar{\ell},\ell)} \mathcal{O}_{(\bar{\ell},\ell)} \rangle(\omega, k=0) \simeq \frac{\beta^{d-2\Delta_{\mathcal{O}}} b_{\bar{\ell}-2}^2}{b_T^2} \frac{b_{\bar{\ell}-2}^2}{\omega} \left(\frac{\beta \omega}{\eta_o}\right)^{\alpha_{\bar{\ell}}}, \tag{3.13}$$

with $\alpha_{\bar{\ell}} = \frac{d}{2} + \bar{\ell} - 2$ for $\bar{\ell}$ even, and $\alpha_{\bar{\ell}} = d + \bar{\ell} - 3$ for $\bar{\ell}$ odd (see (2.17)). Converting this into an expression for the OPE $C_{H_J H'_{J'} \mathcal{O}_\ell}$ using (3.1) and (3.2), we see that the hydrodynamic answer predicts a tensor structure (and fixes the correspondig OPE coefficient) for each $\bar{\ell} = 0, 1, \ldots, \ell$.

---

[†]A CFT is expected to be weakly coupled when its twist gap $\gamma \equiv \min(\Delta - J) - d + 1 \geq 0$ is small $\gamma \ll 1$. In this case the thermalization time is parametrically enhanced $\tau_{\text{th}} \sim \beta/\gamma$, as can be observed e.g. in the $O(N)$ model by comparing its thermalization time [68] to its twist gap [69]. Generic CFTs are expected to satisfy $\gamma \gtrsim 1$ and $\tau_{\text{th}} \sim \beta$.

However if the heavy operators are both scalars, conformal invariance constrains the three-point function up to a single OPE coefficient (the illegal step was to apply ETH (3.1) before accounting for all symmetries). To accommodate the tensor structures obtained from hydrodynamics, it is sufficient to let one of the heavy operators have spin $J \geq \ell$ – this leads to precisely $\bar{\ell} + 1$ tensor structures in agreement with the CFT prediction[†]

$$\langle H, Jm|\mathcal{O}_{\mu_1 \cdots \mu_\ell}|H'\rangle = \sum_{\bar{\ell}=0}^{\ell} C_{H_J H' \mathcal{O}_\ell}^{\bar{\ell}} \delta_{|m|}^{\bar{\ell}} \delta_{\mu_1}^0 \cdots \delta_{\mu_{\ell-\bar{\ell}}}^0 \delta_{\mu_{\ell-\bar{\ell}+1}}^\sigma \cdots \delta_{\mu_\ell}^\sigma + \text{perm} - \text{traces}, \qquad (3.14)$$

where $\sigma = \text{sgn}(m) = \pm$ denotes the spatial directions $\pm = x_1 \pm i x_2$ ($x_1, x_2$ are the directions used to define the weight $m$, i.e. $J_{12}|H, Jm\rangle = |H, Jm\rangle m$).

Combining Eqs. (3.1), (3.2) and (3.13) therefore gives

$$|C_{H_J H_{J'} \mathcal{O}_\ell}^{\bar{\ell}}|^2 \simeq e^{-S} \frac{b_{\bar{\ell}-2}^2}{b_T^2} \frac{(R/\beta)^{\Delta_\mathcal{O}}}{\beta \omega} \left(\frac{\beta \omega}{\eta_o}\right)^{\alpha_{\bar{\ell}}}. \qquad (3.15)$$

Here we have replaced the random number with unit variance $|R_{HH'}|^2 \to 1$. Strictly speaking this expression for $|C_{HH'L}|^2$ and those below should be thought as average statements, averaged over a few heavy operators $H$ or $H'$. We have taken the density of states at energy $E = \Delta/R$ to be

$$\Omega(E) \simeq \beta^{d+1} e^S, \qquad \text{with} \quad S = b_T S_d (R/\beta)^d = b_T S_d \left(\frac{d+1}{dS_d} \frac{\Delta}{b_T}\right)^{\frac{d}{d+1}}. \qquad (3.16)$$

Eliminating the radius $R$ in (3.15) gives (dropping numerical factors)

$$|C_{H_J H_{J'} \mathcal{O}_\ell}^{\bar{\ell}}|^2 \simeq e^{-S} \frac{b_{\bar{\ell}-2}^2}{b_T^2} \left(\frac{\Delta}{b_T}\right)^{\frac{2(\Delta_\mathcal{O}-\alpha_{\bar{\ell}}+1)}{d+1}} \frac{(\Delta-\Delta')^{\alpha_{\bar{\ell}}-1}}{\eta_o^{\alpha_{\bar{\ell}}}} \times \left[1 + O(\omega \tau_{\text{th}}) + O\left(\frac{1}{\omega R}\right)\right]. \qquad (3.17)$$

We will not attempt to control subleading extensions to the Cardy formula (3.16), and therefore will not comment on the subexponential dependence on $\Delta$. However, we attract the reader's attention to the non-analytic dependence on $\Delta - \Delta'$, coming from hydrodynamic fluctuations. Corrections to the leading result are shown in the square brackets, and come from less relevant terms in hydrodynamics and finite volume corrections:

$$\omega \tau_{\text{th}} \sim \frac{\Delta - \Delta'}{(\Delta/b_T)^{\frac{1}{d+1}}}, \qquad \frac{1}{\omega R} \sim \frac{1}{(\Delta - \Delta')(\Delta/b_T)^{\frac{1}{d+1}}}, \qquad (3.18)$$

both of which are parametrically small in the regime (3.11).

If both $J, J' \geq 1$, there may be more tensor structures allowed by conformal invariance than needed – the claim of thermality is that as in (3.8) the leading OPE coefficients will not depend on these extra indices.

### 3.2.2 Mesoscopic spin $J, J'$

Let us now extend to heavy operators with 'mesoscopic' spin. More precisely, we want the spins to be non-macroscopic (so that the state on the sphere remains homogeneous), and the difference in spins to be mesoscopic, or

$$J, J' = o(R^{d+1}), \qquad J - J' = kR, \qquad (3.19)$$

---

[†]In the notation of Ref. [67], the tensor structures in the sum are $H_{\mathcal{O}H}^{\bar{\ell}} V_{\mathcal{O}}^{\ell-\bar{\ell}}$.

in the limit $R \to \infty$. Let us first consider a light operator $\mathcal{O}$ with spin $\ell = 0$. In Sec. 2 we found that its thermal expectation value leads to (see (2.13a))

$$\langle \mathcal{O} \rangle_\beta = \frac{b_0}{\beta^{\Delta_\mathcal{O}}} \quad \Rightarrow \quad \langle \mathcal{O}\mathcal{O} \rangle(\omega, k) \simeq \left( \frac{b_0 \Delta_\mathcal{O}}{\beta^{\Delta_\mathcal{O}}} \right)^2 \frac{2\beta^d}{b_T} \frac{\frac{2d-1}{d^3} \eta_o \beta k^4}{\left( \omega^2 - \frac{1}{d} k^2 \right)^2 + \left( \frac{2d-1}{d} \eta_o \beta \omega k^2 \right)^2} . \tag{3.20}$$

Conformal invariance allows for many tensor structures for the three-point function

$$\langle H, Jm | \mathcal{O} | H', J'm' \rangle = \delta_{mm'} C^{|m|}_{H_J H_{J'} \mathcal{O}} , \tag{3.21}$$

i.e. there is an OPE coefficient for every $|m| = 0, 1, \ldots, \min(J, J')$ (the OPE is diagonal in the weights $m$, $m'$ because a scalar operator $\mathcal{O}$ inserted at the north pole preserves rotations about the pole). However we see from (3.20) that these coefficients do not depend on $m$ and are given by

$$|C^{|m|}_{H_J H_{J'} \mathcal{O}}|^2 \simeq \frac{\alpha}{e^S} \frac{\eta_o (J - J')^4}{\left[ (\Delta - \Delta')^2 - \frac{1}{d}(J - J')^2 \right]^2 + a_d \, \eta_o^2 \left( \frac{b_T}{\Delta} \right)^{\frac{2}{d+1}} (\Delta - \Delta')^2 (J - J')^4} , \tag{3.22}$$

with $a_d = \left( \frac{2(d-1)}{d} \right)^2 \left( \frac{d S_d}{d+1} \right)^{\frac{2}{d+1}}$ and where the subexponential dependence on $\Delta$ (which is degenerate with logarithmic corrections to $S(\Delta)$) was packaged in $\alpha \propto \frac{(b_0 \Delta_\mathcal{O})^2}{b_T^2} \left( \frac{\Delta}{b_T} \right)^{2\Delta_\mathcal{O}/(d+1)}$. These OPE coefficients feature a 'resonance' at the sound mode $\Delta - \Delta' = \pm \frac{1}{\sqrt{d}}(J - J')$. The resonance is sharp for heavy operators, with a width $\frac{\eta_o}{(\Delta/b_T)^{1/(d+1)}} \ll 1$ controlled by the shear viscosity to entropy ratio $\eta_o \equiv \eta/s$. The case $J = J'$ is somewhat special: for this case only the contribution (3.22) vanishes, and the OPE is given by a subleading hydrodynamic tail $|C^{|m|}_{HH'\mathcal{O}}|^2 \sim \alpha e^{-S} (\Delta - \Delta')^{\frac{d}{2}-1}$ similar to (3.17), see appendix A.3.

We are now ready to turn to the general case of the heavy-heavy-light OPE coefficient of three spinning operators. The hydrodynamic prediction was given in Eq. (2.15). To match with OPE coefficients we will need the precise index structure, which can be conveniently packaged by using the index-free notation

$$\langle \mathcal{O}_{(\bar{\ell}, \ell)} \mathcal{O}_{(\bar{\ell}, \ell)} \rangle \equiv z^{i_1} \cdots z^{i_\ell} \langle \mathcal{O}_{i_1 \cdots i_{\bar{\ell}} 0 \cdots 0} \mathcal{O}_{j_1 \cdots j_{\bar{\ell}} 0 \cdots 0} \rangle z'^{j_1} \cdots z'^{j_\ell} , \tag{3.23}$$

with $z^2 = z'^2 = 0$. In this notation, the full index structure of (2.15) is given in appendix A (see Eq. (A.9)). For a CFT (A.9) becomes

$$\begin{aligned} \frac{\langle \mathcal{O}_{(\bar{\ell}, \ell)} \mathcal{O}_{(\bar{\ell}, \ell)} \rangle(\omega, k)}{\beta^{d - 2\Delta_\mathcal{O} + 1}} &= \frac{(b_{\bar{\ell}-1} + b_{\bar{\ell}} \frac{k^2}{\omega\sqrt{d}})^2}{b_T} \frac{\eta_o \omega^2 (k \cdot z)^{\bar{\ell}} (k \cdot z')^{\bar{\ell}}}{\left( \omega^2 - \frac{1}{d} k^2 \right)^2 + \left( \frac{2(d-1)}{d} \right)^2 \eta_o^2 \omega^2 k^4} \\ &+ \frac{(b_{\bar{\ell}-1})^2}{b_T} \left( z \cdot z' - \frac{(k \cdot z)(k \cdot z')}{k^2} \right) \frac{\eta_o k^2 (k \cdot z)^{\bar{\ell}-1} (k \cdot z')^{\bar{\ell}-1}}{\omega^2 + \eta_o^2 k^4} \\ &+ \frac{(b_{\bar{\ell}-2})^2}{b_T^2} \frac{(z \cdot z')^{\bar{\ell}}}{\omega} \left( \frac{\omega}{\eta_o} \right)^{\alpha_{\bar{\ell}}} + \cdots , \end{aligned} \tag{3.24}$$

where we absorbed numerical factors in the coefficients $b_i$, and $\omega$ and $k$ are measured in units of temperature to simplify the expression. The hydrodynamic result (3.24) contains a structure for each $\bar{\ell} = 0, 1, \ldots, \ell$. Moreover, the index contractions in (3.24) take the form

$$(k \cdot z)^r (k \cdot z')^r (z \cdot z')^{\bar{\ell}-r} , \tag{3.25}$$

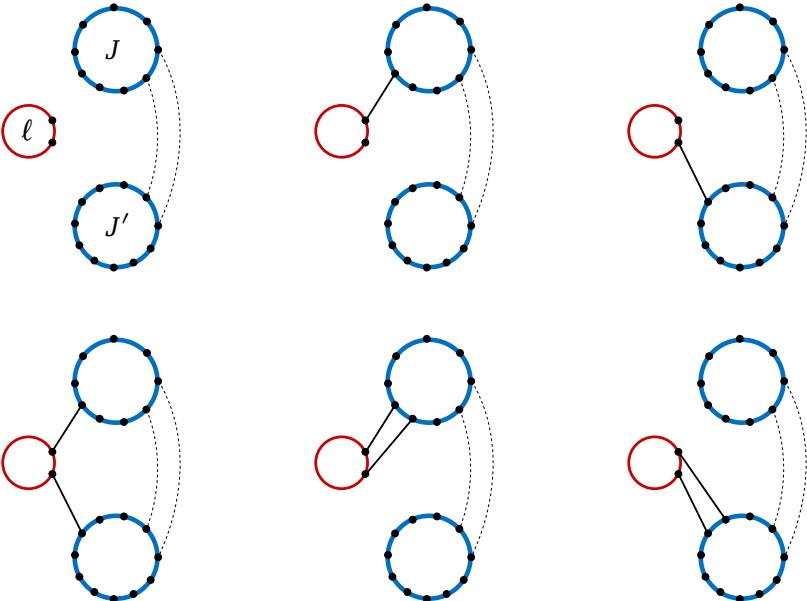

Figure 6: CFT tensor structures for the three point function $\langle H, J | \mathcal{O}_\ell | H', J' \rangle$ with $\ell = 2$, adapted from [67], Fig. 2. The OPEs obtained from hydrodynamics only depend on the $\frac{1}{2}(\ell+1)(\ell+2)$ 'contractions' involving the light operator (solid lines), and not on the contractions between the heavy operators that create the thermal state (dashed lines).

with $r = 0, 1, \ldots, \bar{\ell}$. The leading hydrodynamic result (3.24) only contains these structures for $r = \bar{\ell}, \bar{\ell}-1$ and 0; coefficients of the structures for other values of $r$ will be controlled by subleading hydrodynamic tails (in $d = 2$ these additional tails are only log suppressed compared to the leading ones, see Sec. 2.2). One therefore obtains $\sum_{\bar{\ell}=0}^{\ell}(\bar{\ell}+1) = \frac{1}{2}(\ell+1)(\ell+2)$ structures. When $J, J' \geq \ell$, Ref. [67] showed that the CFT three-point $\langle H | \mathcal{O} | H' \rangle$ function contains more structures: there are $\frac{1}{2}(\ell+1)(\ell+2)\left(\min(J,J')+1-\frac{\ell}{3}\right)$ structures, which in their notation take the form

$$H_{\mathcal{O}H}^a H_{\mathcal{O}H'}^{a'} H_{HH'}^b V_{\mathcal{O}}^{\ell-a-a'} V_H^{J-a-b} V_{H'}^{J'-a'-b}, \tag{3.26}$$

where $a, a', b$ run over all integers such that the powers above are positive. The hydrodynamic OPE coefficients (3.24) only depend on the contractions between the light operator and the heavy ones, hence on $a, a'$ but not on $b$ (see Fig. 6). Since $a, a' = 0, 1, \ldots, \ell$ satisfy $a + a' \leq \ell$ this produces indeed $\frac{1}{2}(\ell+1)(\ell+2)$ structures. We will not explicitly write the map $(a, a') \leftrightarrow (\bar{\ell}, r)$ between the bases (3.25) and (3.26), and instead label OPE coefficients with $\bar{\ell}, r$ and $b$ as $C_{H_J H'_{J'} \mathcal{O}_\ell}^{(\bar{\ell}, r, b)}$. From (3.24) one then finds

$$C_{H_J H'_{J'} \mathcal{O}_\ell}^{(\bar{\ell}, 0, b)} = \text{Eq. (3.17)},$$

$$C_{H_J H'_{J'} \mathcal{O}_\ell}^{(\bar{\ell}, \bar{\ell}-1, b)} \simeq \frac{\alpha}{e^S} \frac{\frac{(b_{\bar{\ell}-1})^2}{b_T} \eta_o (J-J')^{2\bar{\ell}}}{(\Delta - \Delta')^2 + \tilde{a}_d \, \eta_o^2 \left(\frac{b_T}{\Delta}\right)^{\frac{2}{d+1}} (J-J')^4}, \tag{3.27}$$

$$C_{H_J H'_{J'} \mathcal{O}_\ell}^{(\bar{\ell}, \bar{\ell}, b)} \simeq \frac{\alpha}{e^S} \frac{\frac{1}{b_T}\left(b_{\bar{\ell}-1} + b_{\bar{\ell}} \sqrt{\frac{\tilde{a}_d}{d}} \left(\frac{b_T}{\Delta}\right)^{\frac{1}{d+1}} \frac{(J-J')^2}{(\Delta-\Delta')}\right)^2 \eta_o (\Delta-\Delta')^2 (J-J')^{2\bar{\ell}}}{\left[(\Delta-\Delta')^2 - \frac{1}{d}(J-J')^2\right]^2 + a_d \, \eta_o^2 \left(\frac{b_T}{\Delta}\right)^{\frac{2}{d+1}} (\Delta-\Delta')^2 (J-J')^4} - C_{H_J H'_{J'} \mathcal{O}_\ell}^{(\bar{\ell}, \bar{\ell}-1, b)},$$

with $\alpha \propto \left(\frac{\Delta}{b_T}\right)^{\frac{2(\Delta_{\mathcal{O}}-\ell+1)}{d+1}}$ and with the numerical factors $a_d = \left(\frac{2(d-1)}{d}\right)^2 \left(\frac{dS_d}{d+1}\right)^{\frac{2}{d+1}}$, $\tilde{a}_d = \left(\frac{dS_d}{d+1}\right)^{\frac{2}{d+1}}$.
Subleading corrections to these results are similar to those in Eq. (3.17).

### 3.3 Macroscopic spin

Let us now briefly comment on heavy operators with macroscopic spin

$$J \sim \Delta \sim R^{d+1}. \tag{3.28}$$

Macroscopic spin has been treated in an EFT approach for large charge in CFTs with a $U(1)$ symmetry [9, 22]. It was found there that in the regime (3.28), the superfluid state forms a vortex lattice, such that the coarse-grained superfluid velocity is equal to that of a rotating body with angular momentum $J$. For a normal fluid, one expects a similar stationary solution to the Navier-Stokes equations[†]. Let us work in $d = 2$ spatial dimensions for simplicity, and search for a velocity profile $u_\mu \equiv (u_0, u_\theta, u_\phi) = \left((1+v_\phi^2)^{1/2}, 0, v_\phi\right)$ with an azimuthal velocity that only depends on the polar angle $v_\phi = v_\phi(\theta)$. Now a typical state with angular momentum on the sphere will equilibrate (preserving its angular momentum) to an equilibrium velocity profile $v_\phi(\theta)$ that does not dissipate; in particular it must be annihilated by the shear viscosity term in (2.4) – this leads to a differential equation which can be solved for $v_\phi(\theta)$. Energy eigenstates created by heavy operators are expected to look thermal and should have this velocity profile. The ideal stress tensor can then be obtained by imposing conservation

$$T_{\mu\nu} \simeq s_o(\theta)\left(u_\mu(\theta)u_\nu(\theta) + \frac{g_{\mu\nu}}{d+1}\right), \qquad \nabla_\mu T^{\mu\nu} = 0, \tag{3.29}$$

where $g_{\mu\nu}$ is the metric on the sphere. This equation can be solved for $s_o(\theta)$. Finally, computing the total angular momentum of this flow one finds that it is related to the velocity at the equator by

$$v_{\max} = v_\phi(\pi/2) \sim \frac{J}{\Delta}. \tag{3.30}$$

OPE coefficients between heavy operators of macroscopic spin $J$ and light operators can be obtained as in the previous sections by now expanding the constitutive relations around the velocity profile $u_\mu(\theta)$. This hydrodynamic picture is expected to break down near the unitarity bound $J \leq \Delta - d + 1$ – in particular at low twist $\Delta - J \sim 1$ the spectrum is sparse and populated by double- and higher-twist primaries [1,2], see Fig. 2. Increasing twist to go away from the edges of the spectrum will increase the density of state, eventually leading to a finite entropy density and temperature. It is tempting to view the thermal state with macroscopic spin (3.28) as a 'gas of multi-twist states', analogously to how heating up a superfluid leads to a normal fluid component carried by a gas of phonons (this two-fluid picture, and the emergence of dissipative hydrodynamics from a conformal superfluid is discussed in Sec. 4.2.1). The operator phase diagram, including spin, is discussed in more depth in Sec. 4.4 for theories with an additional $U(1)$ symmetry. We leave the study of OPE coefficients for heavy operators with macroscopic spin using hydrodynamics in a rotating background for future work.

## 4 Global symmetries

It is straightforward to extend the results above to QFTs and CFTs with an internal symmetry group $G$; this section deals with the simplest example $G = U(1)$. The additional Ward identity

---

[†]We thank João Penedones for suggesting this.

$\partial_\mu J^\mu = 0$ protects a new slow excitation – charge density $J_0$ – whose fluctuations will give additional contributions to late time correlators.

A background chemical potential $\mu$ can be introduced for the internal symmetry. In sections 4.1 and 4.1.1 we briefly review the hydrodynamic treatment with $\mu = 0$ and $\mu \neq 0$ (a more complete exposition can be found in Ref. [24]) and derive the universal late time behavior of thermal correlators for QFTs with a global $U(1)$ symmetry.

The internal symmetry can be spontaneously broken, in which case the theory is described by dissipative superfluid hydrodynamics[†]. A new feature in this phase is that the late time correlators of operators charged under the $U(1)$ symmetry are also controlled by hydrodynamics, because of the additional hydrodynamic field $\phi$ which non-linearly realizes the $U(1)$ symmetry. In section 4.2, the hydrodynamic treatment of Refs. [71, 72] is reviewed, and late time correlators of light operators derived. In the simplest situations we expect the dissipative superfluids to be smoothly connected to $T = 0$ superfluids on the edge of the spectrum in Fig. 2. This will allow us to connect to recent work on the large charge limit of CFTs [3, 6–9, 22]. The large charge limit can be thought of as a situation where a semiclassical description survives as $T \to 0$ (with fixed $\mu \neq 0$), thanks to spontaneous breaking of the $U(1)$ symmetry.

The implications of long-time tails on the CFT data of CFTs with a $U(1)$ symmetry are studied in section 4.3. Various regions in the $(\Delta, Q)$ plane will be described by the hydrodynamic theories of sections 4.1, 4.1.1 and 4.2, following Fig. 2. Finally, the presence of distinct phases in the large $\Delta$ spectrum naturally brings us to phase transitions. In section 4.4, we study signatures of thermal phase transitions on the CFT data.

## 4.1 Hydrodynamics of a charged fluid

The conservation laws $\partial_\mu T^{\mu\nu} = 0$, $\partial_\mu J^\mu = 0$ must be supplemented with constitutive relations for the currents. In the Landau frame and up to first order in derivatives, the constitutive relation for the stress-tensor is still given by Eq. (2.4) and that of the $U(1)$ current is [24]

$$J^\mu = \rho u^\mu - \kappa \Delta^{\mu\nu} \partial_\nu(\beta\mu) + \chi_T \Delta^{\mu\nu} \partial_\nu \beta + O(\partial^2). \tag{4.1}$$

Three new parameters were introduced: $\rho$, $\kappa$ and $\chi_T$.[‡] These, along with those appearing in (2.4), are functions of both $\mu$ and $\beta$. Consistency with thermodynamics fixes $\rho$ in terms of the equation of state $\rho = \partial P/\partial \mu$, and imposes $\kappa \geq 0$ and $\chi_T = 0$.[§]

Hydrodynamic correlators can again be obtained by expanding around equilibrium (2.5) with $\mu(x) = \mu + \delta\mu(x)$. If we first take the background chemical potential to vanish $\mu = 0$, then the background charge density $\rho$ vanishes by CPT and we see directly from (4.1) that there is no mixing at the linear level between the new hydrodynamic degree of freedom $\delta\mu$ and the ones considered previously $\delta u_\mu$, $\delta\beta$, at least to this order in derivatives. The stress-tensor correlator (2.6) is therefore unchanged, and the current correlator is given by

$$G^R_{J_0 J_0}(\omega, k) = \frac{\chi D_c k^2}{-i\omega + D_c k^2} + \cdots, \tag{4.2}$$

where $\chi \equiv \partial\rho/\partial\mu$ is the charge susceptibility, and $D_c \equiv \kappa\beta/\chi$ the charge diffusion constant.

The late time thermal correlation functions of light operators $\mathcal{O}_\ell$ of spin $\ell$ can be found by matching them to composite hydrodynamic operators as in Section 2. The new hydrodynamic

---

[†]Dissipative superfluid hydrodynamics also describes 2+1d theories at finite temperatures $0 < T < T_{\mathrm{BKT}}$, where strictly there is no spontaneous symmetry breaking; the protection of the long-lived superfluid phase can however be understood without reference to symmetry breaking [70].

[‡]Another commonly used notation for the conductivity is $\sigma_Q \equiv \kappa\beta$.

[§]Note that $\chi_T$ is only forbidden because $J_\mu$ is conserved. Generic non-conserved spin-1 operators will have terms like $\chi_T$ in their constitutive relation. In holographic models, along the lines of Ref. [44], these could come from coupling a massive gauge field in the bulk to the Weyl tensor, e.g. through $A^\mu \partial_\mu C^2_{\mathrm{Weyl}}$.

degree of freedom $\mu$ can now also be used. Comparing (4.2) with (2.6) shows that it scales like the other hydrodynamic fluctuations

$$\delta\mu \sim \delta\beta \sim \delta u_\mu \sim k^{d/2}. \tag{4.3}$$

It is easy to see that the new hydrodynamic field $\delta\mu$ does not allow the construction of more relevant operators – the results from Section 2 are therefore largely unchanged – except for odd spin $\ell$ operators with $\bar{\ell} = 0$ or 1 spatial indices. The reason is that for these cases we found in Sec. 2 that the dominant hydrodynamic contributions to the correlators (2.13) involve the term $\lambda_0$ in (2.10), which was forbidden by CPT for odd-spin operators (see appendix A.2.1). However, thanks to the conserved $U(1)$ charge this term is now allowed for odd spin $\ell$ as well

$$\mathcal{O}_{\mu_1 \cdots \mu_\ell} = \lambda_0(\mu, \beta) u_{\mu_1} \cdots u_{\mu_\ell} + O(\partial), \tag{4.4}$$

where $\lambda_0(\mu, \beta)$ is an odd function of $\mu$ by CPT. Expanding $\lambda_0$ in $\delta\mu$, one finds that components with $\bar{\ell} = 0$ spatial indices can overlap linearly with the density, so that (4.2) implies

$$\langle \mathcal{O}_{0\cdots0} \mathcal{O}_{0\cdots0} \rangle(\omega, k) = \frac{2(\partial\lambda_0/\partial\mu)^2}{\chi\beta} \frac{D_c k^2}{\omega^2 + (D_c k^2)^2} + \cdots, \tag{4.5}$$

and components with $\bar{\ell} = 1$ spatial indices are controlled by a hydrodynamic loop at $k = 0$

$$\langle \mathcal{O}_{i0\cdots0} \mathcal{O}_{j0\cdots0} \rangle(t, k = 0) = \delta_{ij} \frac{(\partial\lambda_0/\partial\mu)^2}{\chi s \beta} \frac{d-1}{d} \frac{1}{[4\pi(D + D_c)t]^{d/2}} + \cdots. \tag{4.6}$$

A special case is the correlator of the current operator itself $\mathcal{O}_\mu = j_\mu$. Then $\lambda_0 = \rho$ so $\partial\lambda_0/\partial\mu = \chi$ and (4.6) reproduces known results [28, 29]. This correlator with $\mathcal{O}_\mu = j_\mu$ is the one that led to the original discovery of long-time tails [33].

### 4.1.1 Turning on a background $\mu \neq 0$

A background chemical potential will allow the longitudinal hydrodynamic modes $j_0$, $T_{00}$ and $\partial_i T_{0i}$ to mix (the transverse sector is unaffected and still given by the second term in (2.6)). The longitudinal sector will still contain a diffusive mode and a sound mode, but these will be carried by linear combinations of $j_0$ and $T_{00}$, see e.g. [24]. The correlators (2.15) of neutral operators therefore do not change qualitatively: the functional dependence on $\omega, k$ is unchanged, but the thermodynamic and transport factors are more complicated.

One exception is again for operators of odd spin $\ell$. For example components with $\bar{\ell} = 1$ spatial indices can now overlap linearly with hydrodynamic modes, even at $k = 0$. Indeed, $\lambda(\mu, \beta)$ in (4.4) is now expanded around $\mu \neq 0$ so that the constitutive relation $\mathcal{O}_{i0\cdots0} = \lambda_0 u_i + \cdots$ has the same form as (2.12b), and the two-point function $\langle \mathcal{O}_{i0\cdots0} \mathcal{O}_{j0\cdots0} \rangle(\omega, k)$ is given by (2.13b). More generally, in charged hydrodynamics at finite density, the results of Sec. 2 hold for both even and odd spin $\ell$, because of the absence of any CPT constraint.

## 4.2 Dissipative superfluids

The hydrodynamic theory of relativistic, dissipative superfluids was thoroughly studied in Refs. [71, 72]. Compared to normal charged fluids, superfluids contain an additional slow hydrodynamic degree of freedom carried by the Goldstone field $\phi$ that non-linearly realizes the internal $U(1)$ symmetry. Here we will focus on conformal superfluids, and will not give an expectation value to the superfluid velocity, i.e. $\langle \partial_i \phi \rangle = 0$. This velocity can be thought of as the charge density associated with an emergent higher-form symmetry [70] – since the symmetry is emergent, heavy CFT operators creating superfluid states are not labeled by their

representations under it. Working to linear order in the superfluid velocity, there is only one new thermodynamic parameter compared to (4.1) – the superfluid stiffness $\rho_s$ – and one new dissipative parameter $\zeta_3$ [71] (see also [72]) :

$$T_{\mu\nu} = \epsilon u_\mu u_\nu + P\Delta_{\mu\nu} + 2\rho_s\mu_s n_{(\mu}u_{\nu)} + \rho_s\mu_s n_\mu n_\nu - \eta\sigma_{\mu\nu} + \cdots , \tag{4.7a}$$

$$J_\mu = \rho_s\partial_\mu\phi + \rho_n u_\mu - \kappa\Delta_{\mu\nu}\partial^{\,\nu}\frac{\mu}{T} + \cdots , \tag{4.7b}$$

$$u^\mu\partial_\mu\phi = -\mu + \zeta_3\partial_\mu(\rho_s n^\mu) + \cdots , \tag{4.7c}$$

where $n_\mu \equiv \Delta_{\mu\nu}\partial^{\,\nu}\phi/\mu_s$ and $\mu_s \equiv -u^\mu\partial_\mu\phi$. The projection $\Delta_{\mu\nu} \equiv \eta_{\mu\nu} + u_\mu u_\nu$, and $\sigma_{\mu\nu}$ is the shear viscosity tensor appearing in (2.4). The thermodynamic parameters $\epsilon$, $P$, $\rho_s$, $\rho_n$ and dissipative parameters $\eta$, $\kappa$, $\zeta_3$ are inputs in the hydrodynamic treatment – however when the dissipative superfluid is obtained by heating up a $T = 0$ conformal superfluid (as in Fig. 2), these can all be expressed in terms of a single EFT parameter at low temperature, see section 4.2.1.

The fluctuations $\partial_\mu\delta\phi$, with $\delta\phi = \phi + \mu t$, have the same scaling as the other hydrodynamic variables (4.3). This new degree of freedom lifts the diffusive mode (4.2) into a second sound mode (with sound attenuation controlled by $\kappa$ and $\zeta_3$). The correlation function for $\partial_i\delta\phi$ is qualitatively similar to the longitudinal part of (2.6), and correlators of neutral operators will be controlled by similar hydrodynamic tails as in the previous sections.

One new feature is that operators with finite charge $q \in \mathbb{Z}$ under the $U(1)$ symmetry can now be matched in the IR using the Goldstone phase

$$\mathcal{O}^q_{\mu_1\cdots\mu_\ell} \sim \partial_{\mu_1}\cdots\partial_{\mu_\ell}e^{iq\phi} + e^{iq\phi}u_{\mu_1}\partial_{\mu_2}\cdots\partial_{\mu_{\ell-1}}u_{\mu_\ell} + \cdots , \tag{4.8}$$

where as in Fig. 1 the first term is the most relevant operator, and the second is the most relevant operators when $k = 0$. There are several operators that compete with the ones above, but all lead to similar results. The correlators of charged operators $\mathcal{O}^q$ can now be obtained as those of neutral operators $\mathcal{O}$ in Sec. 2.1, by expanding the hydrodynamic fields. Expanding $\phi = \delta\phi - \mu t$ shows that real time correlators contain an extra factor of $e^{-iq\mu t}$, so that in frequency space one finds

$$\langle \mathcal{O}^{q\dagger}_{(\bar\ell,\ell)}\mathcal{O}^q_{(\bar\ell,\ell)}\rangle(\omega,k) \sim \langle \mathcal{O}_{(\bar\ell,\ell)}\mathcal{O}_{(\bar\ell,\ell)}\rangle(\omega - q\mu, k), \tag{4.9}$$

where the right-hand side simply refers to the general result (2.15) for neutral operators, evaluated at frequency $\omega - q\mu$. One may worry that the hydrodynamic features in this correlator appear at frequencies above the hydrodynamic cutoff $\omega \sim q\mu \gg 1/\tau_{th}$ – however this is simply because the operator carries a phase $e^{-iq\mu t}$ which translates hydrodynamic features usually at $\omega \sim 0$ to $\omega \sim q\mu$ in the correlator of operators of charge $q$. We will see in the CFT application in Sec. 4.3 that $\omega - q\mu$ measures the difference in dimensions $\Delta - \Delta'$ of the heavy operators (as did $\omega$ for neutral operators, see (3.9)).

Of course, superfluids at finite temperature also have well known static properties which control equal time correlators. For example terms like the first term in (4.8) will lead to

$$\langle \mathcal{O}^{q\dagger}_{(\bar\ell,\ell)}\mathcal{O}^q_{(\bar\ell,\ell)}\rangle(x,t=0) \sim e^{-\frac{q^2}{2}\langle\phi(x)\phi\rangle}\partial_i^{2\bar\ell}\langle\phi(x)\phi\rangle \sim \frac{1}{x^{2\bar\ell+d-2}} \qquad \text{for } d > 2. \tag{4.10}$$

For $d = 2$, where the equal-time phase correlator $\langle\phi(x)\phi\rangle = \frac{1}{2\pi\rho_s}\log|x|$ (where $\rho_s$ is the stiffness in (4.7)) one has instead

$$\langle \mathcal{O}^{q\dagger}_{(\bar\ell,\ell)}\mathcal{O}^q_{(\bar\ell,\ell)}\rangle(x,t=0) \sim \frac{1}{x^{2\bar\ell+\frac{q^2}{4\pi\rho_s}}}. \tag{4.11}$$

These also provide EFT constraints on the CFT data involving heavy charged operators that create a superfluid state.

### 4.2.1 Dissipative superfluids from the EFT

It is natural to expect that if a CFT exhibits a superfluid phase, this phase will be connected to a $T = 0$ superfluid, as in Fig. 2. At $T = 0$, a superfluid EFT describes the physics up to a cutoff, which in the case of a CFT must be proportional to the chemical potential $\mu$ [6, 8, 73, 74]. Dissipative hydrodynamics can be seen to emerge from the EFT at finite temperatures $0 < T \ll \mu$; in other words, all the thermodynamic parameters and dissipative parameters in the previous section can be computed in terms of the EFT parameters. Let us illustrate this to leading order in gradients, where the EFT is simply [74]

$$S = \frac{c_1}{d(d+1)} \int d^{d+1}x \, |\partial \phi|^{d+1} + \cdots, \tag{4.12}$$

with $|\partial \phi| \equiv \sqrt{-\partial_\mu \phi \partial^\mu \phi}$. The dimensionless constant $c_1$ is non-universal and depends on the underlying CFT. The $U(1)$ current is

$$J_\mu = \frac{c_1}{d} |\partial \phi|^{d-1} \partial_\mu \phi + \cdots. \tag{4.13}$$

Expanding around the saddle $\phi = \mu t + \pi$, one finds a zero temperature superfluid density

$$\rho_s = \langle J_0 \rangle_{\beta \to \infty} = \frac{c_1}{d} \mu^d. \tag{4.14}$$

Thermodynamics $\delta \epsilon = \mu \delta \rho$ then fixes the zero temperature energy density

$$\epsilon \equiv \langle T_{00} \rangle_{\beta \to \infty} = \frac{c_1}{d+1} \mu^{d+1}, \tag{4.15}$$

as can be checked by computing the stress tensor directly from (4.12). The pressure for a CFT is given by $P = \epsilon/d$. From the CFT perspective, $c_1$ can be defined by Eq. (4.14) or (4.15) and can be viewed as CFT data on a similar footing as the thermal expectation value of the stress tensor, $b_T$ in (3.5).

The action (4.12) can be expanded around the saddle $\phi = \mu t + \pi$

$$S = S_2 + S_{\text{int}}, \qquad S_2 = \int d^{d+1}x \, \frac{1}{2}\left( \dot{\pi}_c^2 - \frac{1}{d}(\nabla \pi_c)^2 \right), \tag{4.16a}$$

$$S_{\text{int}} \sim \int d^{d+1}x \, \frac{(\partial \pi_c)^3}{\sqrt{\epsilon}} + \frac{(\partial \pi_c)^4}{\epsilon} + \cdots. \tag{4.16b}$$

with $\pi_c \equiv \sqrt{c_1 \mu^{d-1}} \, \pi$. Only the schematic form of the interactions $S_{\text{int}}$ will be needed – $\partial \pi_c$ symbolizes either time or space derivatives and numerical factors have been dropped. The strong coupling scale of the EFT is given by the energy density $\Lambda_{\text{sc}} \sim \epsilon^{1/(d+1)} \sim c_1^{1/(d+1)} \mu$.

Let us start with the thermodynamics, which to leading order can be studied from the free part Euclidean version of the action (4.16a). The simplest finite temperature quantity to compute is the entropy density, which can be obtained from the free energy

$$\begin{aligned} f &= -\frac{1}{\beta V} \log Z \simeq -\frac{1}{\beta V} \log \int D\phi \, e^{-S_{2,E}} \\ &= \frac{1}{\beta} \int \frac{d^d k}{(2\pi)^d} \log\left[1 - e^{-c_s \beta k}\right] = \frac{1}{c_s^d \beta^{d+1}} \frac{\Gamma(\frac{d+1}{2})\zeta(d+1)}{\pi^{(d+1)/2}}, \end{aligned} \tag{4.17}$$

with $c_s = 1/\sqrt{d}$, from which we can obtain the dimensionless entropy density

$$s_o^{\text{sflu}} \equiv \beta^d s = -\beta^{d+2}\partial_\beta f \simeq \frac{d+1}{c_s^d} \frac{\Gamma(\frac{d+1}{2})\zeta(d+1)}{\pi^{(d+1)/2}}. \tag{4.18}$$

Terms that are higher order in gradients or field in the action (4.12) and (4.16) lead to corrections to the expressions above that are suppressed by powers of $T/\mu$. The normal density is slightly more subtle: it comes from taking the thermal expectation value of nonlinear terms in the current (4.13), and showing the disalignment between the current and the expectation value of $\partial_\mu \phi$ [70]. One finds

$$\rho_{\mathrm{n}} = \frac{s}{\beta\mu}\frac{1-c_s^2}{c_s^2} + \cdots, \tag{4.19}$$

with $s$ given by (4.18). The non-relativistic limit $c_s \ll 1$ of this expression is well known [75]. A similar expression has appeared in a holographic context recently [76] – we see here that it is a universal prediction of the EFT. For a CFT, $c_s^2 = 1/d$. Furthermore, the emergence of hydrodynamics at finite temperature leads to an additional sound mode (second sound) – since the EFT is to leading order a free scalar and hence scale invariant at low energies, the speed of second sound is itself related to that of first sound as $c_{s,2}^2 = c_s^2/d$ at low temperatures $T \ll \mu$ (see e.g. [75, 77]).

Finally, the dissipative parameters $\eta, \kappa, \zeta_3$ that appeared in the previous section can also be computed from the EFT (4.16) by treating the weakly coupled phonons with kinetic (Boltzmann) theory. This was done for non-conformal superfluids (which have two additional viscosities $\zeta_1, \zeta_2$) in the non-relativistic limit in Ref. [75]. The calculation is quite lengthy so we only sketch it here, focusing on the shear viscosity $\eta$ for illustration. The phonon differential cross section can be computed at tree level from the cubic and quartic terms in (4.16) (see e.g. [78]), the diagrams in Fig. 7 lead to

$$\frac{d\sigma}{d\Omega} \sim \frac{p^{d+3}}{\epsilon^2}, \tag{4.20}$$

where $\epsilon$ is the energy density (4.15) and $p$ symbolizes dependence on the individual phonon momenta $p_i$, $i = 1, 2, 3, 4$. The dependence on the individual momenta can be important, in particular the total cross section $\sigma$ diverges because of small angle scattering [75, 78]. This divergence is regulated by more irrelevant terms in the action (4.12), so that the total cross-section is less suppressed by the cutoff $\epsilon$ than Eq. (4.20) suggests [75]. However it is large angle scattering that controls the shear viscosity [13], so that the naive expression (4.20) is sufficient for our parametric estimate. One can now estimate the thermalization time from the thermally averaged cross-section

$$\tau_{\mathrm{th}} \sim \frac{1}{\langle s\sigma v\rangle} \sim \beta(\epsilon\beta^{d+1})^2. \tag{4.21}$$

The thermalization time is large $\tau_{\mathrm{th}} \gg \beta$ because the phonons are weakly coupled. The shear viscosity can then be estimated as

$$\eta \sim \frac{s\tau_{\mathrm{th}}}{\beta} \sim \epsilon^2\beta^{d+2} \sim c_1^2\mu^{2d+2}\beta^{d+2}. \tag{4.22}$$

The viscosity diverges rapidly as $T \to 0$ because of the long thermalization time (4.21) of the superfluid.

It is interesting to contrast these results to holographic superfluids [79, 80]. Because these theories have a large $O(N^2)$ number of degrees of freedom, the superfluid sector only gives small $O(1)$ corrections to thermodynamic quantities such as the entropy density $s$. However, transport is more sensitive to the presence of the weakly coupled superfluid sector. The holographic value of the low temperature shear viscosity

$$\eta = \frac{s}{4\pi} \sim N^2 T^d \tag{4.23}$$

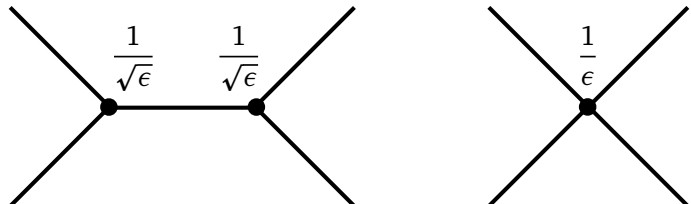

Figure 7: Diagrams in the superfluid EFT contributing to the shear viscosity $\eta$ and other transport parameters $\kappa$, $\zeta_3$ at leading order in $T/\mu$.

should receive a subleading in $N^2$ phonon contribution (4.22), which dominates for temperatures

$$T \lesssim \mu \left(\frac{c_1}{N}\right)^{1/(d+1)}. \tag{4.24}$$

A similar conclusion holds for more general hyperscaling-violating Lifschitz geometries where $\eta = \frac{s}{4\pi} \sim N^2 \mu^d (T/\mu)^{\frac{d-\theta}{z}}$ [76, 81], with a different exponent in (4.24). It would be interesting to understand if this non-commutativity of the $T \to 0$ and $N \to \infty$ limits signals a more important breakdown of low temperature finite density holographic solutions (such as extremal black holes) due to quantum effects [82].

## 4.3 Implications for heavy CFT operators with macroscopic charge

The ETH Ansatz (3.1) is slightly modified for systems with additional symmetries. For the case of an internal $U(1)$ symmetry with generator $\hat{Q}$, the extension can be simply obtained by using the Hamiltonian $\hat{H} \to \hat{H} - \mu\hat{Q}$ without the need of using the grand canonical ensemble explicitly. One then obtains, for a few-body operator $\mathcal{O}_q$ of $U(1)$ charge $q$,

$$\langle H', Q + q|\mathcal{O}_q|H, Q\rangle = \langle \mathcal{O}_q \rangle \delta_q^0 \delta_{HH'} + \Omega(p)^{-1/2} R_{HH'}^{\mathcal{O}} \sqrt{\langle \mathcal{O}_q^\dagger \mathcal{O}_q \rangle (E - E' - \mu q)}. \tag{4.25}$$

Both the one-point and two-point functions are evaluated at finite inverse temperature $\beta$ and chemical potential $\mu$ related to the charge and energy density of $|H, Q\rangle$ by the equation of state. For neutral light operators $q = 0$ the results of section 3 are largely unchanged. One exception is for light operators of spin $\ell = 1$, see Eq. (4.6) and discussion in Sec. 4.1.1; the resulting OPE predictions can be straightforwardly obtained following the method in Sec. 3.

In a superfluid phase, we found in Eq. (4.9) that the correlators of light charged operators $\mathcal{O}_q$ are also controlled by hydrodynamics. Therefore, when the state created by the heavy operator $H_{Q,J}$ is a finite temperature superfluid we can use (4.25) to obtain hydrodynamic predictions for OPE coefficients of light charged operators. We find that the results in Sec. 3 for neutral operators ($q = 0$) are essentially unchanged, but now also hold for charged operators (with the obvious constraint of charge conservation). For example (3.17) becomes

$$|C_{H_{Q,J}H_{Q+q,J'}^{\dagger}\mathcal{O}_{q,\ell}}^{\bar{\ell}}|^2 \simeq e^{-S} \frac{b_{\bar{\ell}-2}^2}{b_T^2} \left(\frac{\Delta}{b_T}\right)^{\frac{2(\Delta_\mathcal{O} - \alpha_{\bar{\ell}} + 1)}{d+1}} \frac{(\Delta - \Delta')^{\alpha_{\bar{\ell}} - 1}}{\tilde{\eta}_o^{\alpha_{\bar{\ell}}}}, \tag{4.26}$$

the only difference with (3.17) being that this also holds for $q \neq 0$ and the relevant transport parameter $\tilde{\eta}_o$ is not simply the shear viscosity but a combination of the superfluid dissipative parameters $\eta$, $\kappa$ and $\zeta_3$ from Sec. 4.2. The other results in Sec. 3 are similarly generalized. For example, for a light charged scalar $\mathcal{O}_q$ a result similar to (3.22) holds: the OPE coefficient features hydrodynamic poles, but there are now two sound modes (first and second superfluid sound), with speed of sound that are no longer fixed to $1/\sqrt{d}$ by conformal invariance.

Further increasing the charge $Q$ of the heavy operator, one eventually reaches the edge of the spectrum. If the operator at the edge of the spectrum still creates a state of finite charge and energy density, its dimension must satisfy

$$\Delta_{\min}(Q) \propto Q^{\frac{d+1}{d}} \,. \tag{4.27}$$

A natural possibility is that this state is a superfluid [6]. The superfluid EFT then predicts both the spectrum of low-lying operators, and OPE coefficients between these operators and light CFT operators [3, 8]. As one moves away from the edge $\Delta \geq \Delta_{\min}(Q)$, the spectrum becomes dense and the many phonon state eventually start to look thermal. Notice that here the thermalization time is large (4.21), because the original EFT is weakly coupled. The dimension of operators near the edge can be written as

$$\Delta = (1 + \delta)\Delta_{\min}(Q) \,, \tag{4.28}$$

where $\delta \ll 1$ is related to the temperature by (this relation follows from Eq. (4.40) derived in the following section)

$$\delta \sim \frac{s_o^{\mathrm{sflu}}}{\epsilon \beta^{d+1}} \,, \tag{4.29}$$

with $s_o^{\mathrm{sflu}}$ given by (4.18). This implies that the hydrodynamic window (3.11) is parametrically smaller close to the edge of the spectrum

$$\left(\frac{\delta}{s_o^{\mathrm{sflu}}}\right)^{-2} \left(\frac{\Delta\delta}{s_o^{\mathrm{sflu}}}\right)^{-\frac{1}{d+1}} \lesssim \Delta - \Delta' \lesssim \left(\frac{\delta}{s_o^{\mathrm{sflu}}}\right)^{2} \left(\frac{\Delta\delta}{s_o^{\mathrm{sflu}}}\right)^{\frac{1}{d+1}} \,. \tag{4.30}$$

## 4.4 Phase transitions in the spectrum

The equation of state of a CFT at finite $\mu$ and $\beta$ is no longer fixed by scale invariance, but can depend on the dimensionless reduced chemical potential

$$\alpha \equiv \beta\mu \,. \tag{4.31}$$

In the previous sections, we explored the hydrodynamic descriptions pertaining to two natural phases of CFTs at finite density – the superfluid phase that is expected for $\alpha \gtrsim 1$ and normal phase for $\alpha \lesssim 1$ – and determined how hydrodynamics controls some of the CFT data. These phases should be separated by a phase transition. In this section, we explore how the non-trivial *thermodynamic* properties of the transition control the data of the underlying CFT, and leave for future work a hydrodynamic treatment of the system near the phase transition (this would require incorporating long-lived critical fluctuations, see e.g. Refs. [83, 84]). In this sense, this section extends the work of Ref. [10], where thermodynamics was seen to control some of the CFT data, to situations where the thermodynamic equation of state and corresponding phase structure are non-trivial.

Expectation values of the currents now take the form

$$\langle J_\mu \rangle_{\beta,\mu} = \frac{\rho_o(\alpha)}{\beta^d} \delta_\mu^0 \,, \qquad \langle T_{\mu\nu} \rangle_{\beta,\mu} = \frac{s_o(\alpha)}{\beta^{d+1}} \left( \delta_\mu^0 \delta_\nu^0 - \mathrm{trace} \right) \,, \tag{4.32}$$

where $\rho_o$ and $s_o$ are odd and even functions of $\alpha$ respectively (by CPT), and $s_o(0) = b_T$. In a CFT, the thermodynamic relations

$$\delta\epsilon = T\delta s + \mu\delta\rho \,, \qquad \frac{d+1}{d}\epsilon = Ts + \mu\rho \,, \tag{4.33}$$

reduce the equation of state to a single function of one variable, which we could take for example to be $s_o(\alpha)$. However when studying the operator spectrum in a CFT, it is most convenient to work in the microcanonical ensemble and to think instead of $\alpha$ (or $\mu$) and $\beta$ as functions of the densities, say $\epsilon$ and $\rho$. In particular, it will be convenient to study a slice of Fig. 2 at fixed $\Delta \gg 1$, i.e. fixed energy density $\epsilon$, and vary charge. Since $\epsilon$ is fixed, we can use it to define a dimensionless charge density and temperature

$$n \equiv \frac{\rho}{\epsilon^{\frac{d}{d+1}}} = \frac{Q}{(S_d \Delta^d)^{\frac{1}{d+1}}}, \qquad \bar{\beta} \equiv \beta \epsilon^{1/(d+1)}, \qquad (4.34)$$

where again $S_d \equiv \text{Vol} S^d = \frac{2\pi^{(d+1)/2}}{\Gamma(\frac{d+1}{2})}$. The potentials $\alpha(n)$ and $\bar{\beta}(n)$ are dimensionless functions of the dimensionless charge density $n$. The thermodynamic relations (4.33) imply that these functions satisfy

$$n\partial_n \alpha(n) = \frac{d+1}{d} \partial_n \bar{\beta}(n), \qquad (4.35)$$

so that only one function is independent, say $\bar{\beta}(n)$, and can be thought of as the equation of state characterizing the thermodynamic properties of the CFT. Thermodynamic stability further implies

$$\partial_n \bar{\beta}(n) \geq 0, \qquad (4.36)$$

so that both $\alpha$ and $\bar{\beta}$ are positive, monotonically increasing functions of $n$.

The asymptotic properties of the equation of state can be related to familiar parameters of the CFT. For example, as $n \to 0$ one has

$$\bar{\beta}(n) = \left( \frac{b_T d}{d+1} \right)^{\frac{1}{d+1}} \left[ 1 + \frac{1}{2} \frac{d}{d+1} \frac{n^2}{\chi_o} + O(n^4) \right] \qquad (\text{as } n \to 0). \qquad (4.37)$$

The first term simply comes from (3.5), and the subleading term follows from (4.33) and (4.35) and features the dimensionless charge susceptibility

$$\chi \equiv \lim_{\mu \to 0} \frac{\langle J_0 \rangle_{\beta, \mu}}{\mu}, \qquad \chi_o \equiv \chi / \epsilon^{\frac{d-1}{d+1}}. \qquad (4.38)$$

($\chi$ can also be expressed as a thermal 2-point function of the current at zero chemical potential). The monotonicity of $\bar{\beta}$ (4.36) for $n \ll 1$ is equivalent to $\chi_o \geq 0$.

The equation of state is also fixed in the opposite limit if we assume, following Ref. [6], that the state at

$$n \to n_{\max} = \frac{Q_{\max}}{(\Delta^d S_d)^{\frac{1}{d+1}}} = \frac{(d+1)^{\frac{d}{d+1}}}{d} c_1^{\frac{1}{d+1}} \qquad (4.39)$$

is a zero-temperature superfluid[†]. Using again the thermodynamic identities (4.35), one finds that the equation of state near the zero-temperature superfluid takes the form

$$\bar{\beta}(n) = \left[ \frac{d}{d+1} \frac{s_o^{\text{sflu}}}{1 - \frac{n}{n_{\max}}} \right]^{\frac{1}{d+1}} + \cdots, \qquad (\text{as } n \to n_{\max}), \qquad (4.40)$$

where $s_o^{\text{sflu}}$ is given by (4.18). Note that the two asymptotic behaviors (4.37) and (4.40) of $\bar{\beta}(n)$ are consistent with its monotonicity property (4.36). A sketch of the equation of state is shown in Fig. 8.

---

[†]This equation can be viewed as a microcanonical CFT definition of the EFT parameter $c_1$. Alternatively Eqs. (4.14) or (4.15) are canonical definitions of $c_1$.

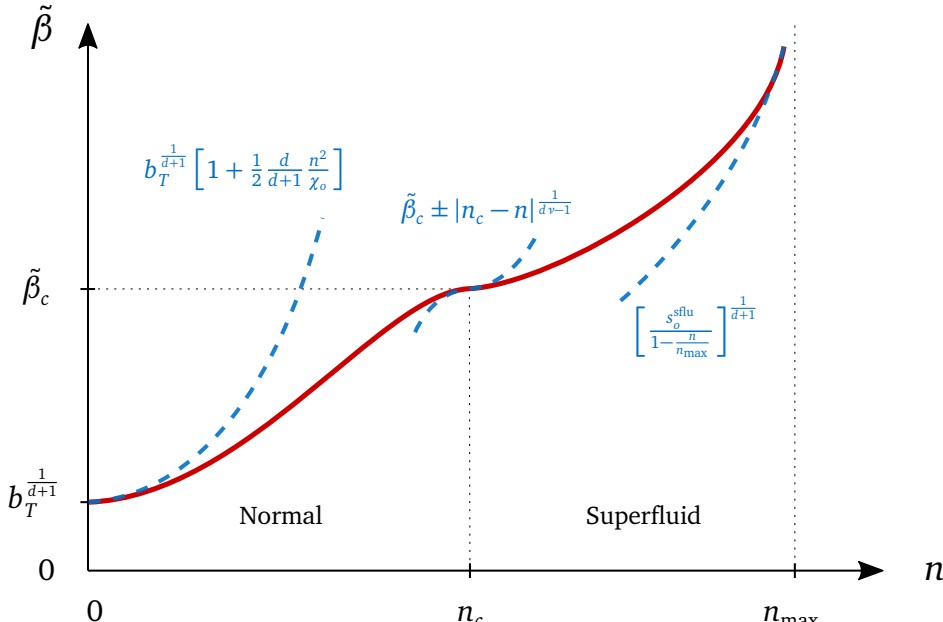

Figure 8: Equation of state for a CFT with a global $U(1)$ symmetry, assuming it reaches a superfluid phase at zero temperature and finite chemical potential. The equation of state (red) can be parametrized by the dependence of the dimensionless inverse temperature $\bar{\beta} = \beta \epsilon^{\frac{1}{d+1}}$ on the dimensionless density $n = \rho/\epsilon^{\frac{d}{d+1}}$ at fixed energy density $\epsilon$, the $y$ axis is normalized as $\tilde{\beta} \equiv \left(\frac{d+1}{d}\right)^{\frac{1}{d+1}} \bar{\beta}$ for convenience. The dashed blue curves show the behavior near $n = 0$ (Eq. (4.37)), $n = n_c$ and $n = n_{\max}$ (Eq. (4.40)).

Now the superfluid phase is certainly not expected to persist at large temperatures $\beta\mu \ll 1$ (or small charge at fixed energy $n \ll 1$)[†]; we therefore expect the symmetry to be restored at a critical value $n = n_c$, with $n_c = O(1)$ for a generic CFT. If this thermal phase transition is continuous, we see that the spectra of $(d+1)$-dimensional CFTs contain information about criticality in $d$ dimensions. Using scaling relations the critical point can be characterized by a correlation length critical exponent $\nu$ and anomalous dimension $\eta$ of the order parameter[‡]. Holographic superfluids are an example of CFTs that can be tuned across a $U(1)$-restoring thermal phase transition[§]. That the transition is in the mean-field universality class in this case [79], with $\eta = 0$ and $\nu = 1/2$, is likely an artefact of large $N$; mean-field critical exponents are not expected for generic CFTs.

Consider for example a $(3+1)d$ CFT with a global $U(1)$ symmetry, and assume following Ref. [6] that the lightest operator of charge $Q$ creates a superfluid state when $n = n_{\max}$. When $n$ is decreased past $n_c$, the symmetry is restored and we expect the transition to be in the 3d Wilson-Fisher universality class, with $\nu \simeq 0.672$ and $\eta \simeq 0.038$[¶]. These exponents control correlators near or at the critical $n_c$, which like the hydrodynamic long-time tails will lead to predictions for some of the CFT data. For example the anomalous dimension $\eta$ will control

---

[†]See however [85] for constructions in fractional dimensions of ordered finite temperature phases at zero density.

[‡]When a $d$-dimensional Euclidean CFT describes the critical point, these are related to the dimensions of the lightest neutral scalar $\Delta_s = d - \frac{1}{\nu}$ and charged order parameter $\Delta_{\vec{\phi}} = \frac{1+\eta}{2}$. Even then we purposely use 'old-fashioned' notation for critical exponents $\nu, \eta$ to avoid confusion with the underlying $(d+1)$-dimensional Lorentzian CFT.

[§]See Fig. 3 in [86] for a distribution of $n_c$ in a class of holographic superfluids.

[¶]See Ref. [87] for a recent discussion on the $8\sigma$ tension between the numerical and experimental values of these exponents.

the equal-time correlator of light, charged operators

$$\langle \mathcal{O}_q(0,x)\mathcal{O}_q^\dagger \rangle_{\beta_c} \sim \frac{1}{x^{d-2+\eta}}. \tag{4.41}$$

The correlation length critical exponent can be obtained from the vanishing of the thermal mass at the critical point. The thermal mass $m_{\text{th}} = m_{\text{th}}(\beta,\mu)$ is defined in the normal (non-superfluid) phase as the decay of spatial correlators of light operators at finite temperature (see e.g. [18])

$$\lim_{x \to \infty} \langle \mathcal{O}_q(0,x)\mathcal{O}_q^\dagger \rangle_\beta \sim e^{-m_{\text{th}}|x|} \tag{4.42}$$

(in the superfluid side $n > n_c$, these correlators decay polynomially, see (4.10)). As we approach the critical point from the normal phase $n \to n_c$, the thermal mass should vanish as

$$m_{\text{th}}(n) \sim |\beta(n) - \beta_c|^\nu. \tag{4.43}$$

Because scaling relations connect several observables, Eqs. (4.41) and (4.43) are but one of several ways to observe the $3d$ critical exponents $\nu$ and $\eta$ in the $(3+1)d$ CFT data. Transitions are only sharp in strict thermodynamic limit $\Delta = \infty$ – thermodynamic singularities are as usual resolved at finite volume, or here finite $\Delta \gg 1$.

The case of $(2+1)d$ CFTs with a $U(1)$ symmetry is particularly interesting. Let us consider the $(2+1)d$ $U(1)$ Wilson-Fisher CFT to be concrete. Monte-Carlo simulations have shown a $\Delta_{\min}(Q) \sim Q^{3/2}$ scaling of the lightest operator at fixed $Q$ [88], implying that this operator creates a state with both finite energy and charge density. Since the theory is fully bosonic, this state is expected to be in a superfluid phase. As $n$ is decreased past $n_c$, the $U(1)$ symmetry is restored – since now $d = 2$ we expect the transition to be in the Berezinskii-Kosterlitz-Thouless (BKT) universality class. In particular the thermal mass in the normal phase near the transition behaves as

$$m_{\text{th}}(n) \sim \exp\left[ -\frac{1}{\sqrt{\beta_c - \beta(n)}} \right], \tag{4.44}$$

and the equation of state $\beta(n)$ is very smooth, with an essential singularity at $n = n_c$.

The phase diagram can be considerable enriched by considering operators with spin $J = j\Delta$, with $0 \le j \le 1$ (still in the $\Delta \to \infty$ limit)[†]. The corresponding states at zero temperature, i.e. keeping the charge density as large as possible $n = n_{\max}(\epsilon, j)$, were studied in $d = 2$ and $d = 3$ in Refs. [9,22], where it was found that the angular momentum of the state is carried by different objects (on top of the superfluid background) depending on $j$:

$$0 \le j \lesssim \Delta^{-\frac{d}{d+1}} \qquad \text{single phonon} \tag{4.45a}$$

$$\Delta^{-\frac{d}{d+1}} \lesssim j \lesssim \Delta^{-\frac{1}{d+1}} \qquad \text{vortex-antivortex pair} \tag{4.45b}$$

$$\Delta^{-\frac{1}{d+1}} \lesssim j \lesssim 1 \qquad \text{vortex crystal} \tag{4.45c}$$

As $j = \frac{J}{\Delta} \to 1$, the superfluid EFT breaks down and the spectrum is instead governed by the light-cone bootstrap. Departing from the manifold of maximal charge at fixed dimension and spin, these 'phases' will be embedded in a larger phase diagram with finite temperature phases. At large enough temperatures, the $U(1)$ symmetry will be restored, and the vortex lattice will melt. In Fig. 9, we show a tentative operator spectrum 'phase diagram' for heavy operators $\Delta \gg 1$ of a CFTs with a global $U(1)$ symmetry.

Similar phase diagrams have been observed in liquid helium [89], Bose-Einstein condensates [90], thin film superconductors [91], and quantum Hall systems; in the last spin per

---

[†]For $d > 2$, the Lorentz group has more than one Cartan generator, but we will only consider one large spin quantum number for simplicity, see [22] for a more general study.

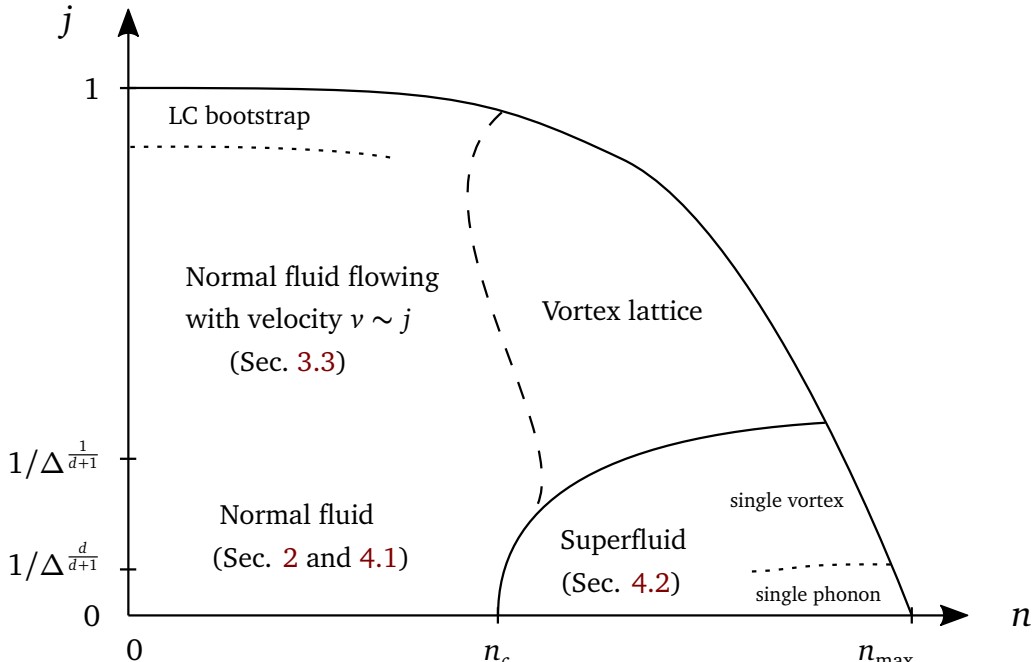

Figure 9: Cut in the spectrum (Fig. 2) at fixed $\Delta \gg 1$, showing a possible 'heavy operator phase diagram' for CFTs with a global $U(1)$ symmetry, as a function of their charge $n \sim Q/\Delta^{\frac{d}{d+1}}$ and spin $j \equiv J/\Delta$. Although certain limiting regions are fairly well understood, most regions, cross-overs and transitions are conjectural. For example, a continuous superfluid to normal transition at $j \ll 1$ could also turn into a first order transition at larger $j$, as is observed in holographic superfluids [80].

charge is mapped to the filling fraction $\nu = J/Q \sim \Delta^{\frac{1}{d+1}} j/n$ (see e.g. [90, 92, 93]). Comparison with these systems suggest a number of possible exotic features in the phase diagram in Fig. 9. For example in (2+1)d, the spinning operators studied in [9] can lead to states with opposite vorticity on each poles, and vanishing vorticity along the equator. Gapless edge states are then expected to live along the equator. Since these are supported in (1+1)d, their hydrodynamic interactions are relevant and dissipation anomalous [25–27, 51]. The CFT spectrum may also probe the melting of the vortex lattice in Fig. 9. In (2+1)d this transition is infinite order like BKT (4.44), but with different exponents [94]. Finally dynamical response and transport near the equilibrium critical point is also singular [83]. We leave a more thorough exploration of this phase diagram for future work.

## 5 Conclusion

We showed that hydrodynamics controls a large portion of the CFT data, namely OPE coefficients of any two heavy operators close enough in dimensions (see (1.5)) with light neutral operators of any spin. Only light operators with internal quantum numbers can escape this fate: for example fermions, or $\mathbb{Z}_2$-odd operators in the Ising model. In superfluid states we found that even light operators that are charged under the $U(1)$ have hydrodynamical OPE coefficients. More generally, when the thermal state created by the heavy operator contains long-lived excitations that nonlinearly realize a global symmetry, hydrodyanmics will control the evolution of light operators charged under that symmetry.

Our results apply to thermalizing CFTs in $d+1$ dimensions with $d \geq 2$. The infinite tower of

Virasoro symmetries make 1+1d CFTs special. In the thermodynamic limit, thermal correlation functions are trivial and there is no room for a hydrodynamic description. However they still exhibit thermalization after a quench [95] (towards a generalized Gibbs ensemble for the KdV charges [96–100]), non-trivial non-equilibrium behavior [101], and chaos [102–104]. It would be interesting to see if out of equilibrium methods can be used to determine heavy-heavy-light OPE coefficients, comparing against results obtained from other methods [105–109], see in particular [110–112] for discussions on the off-diagonal part of ETH in this context. Far from equilibrium techniques and turbulence may also be useful in higher dimensions to determine OPE coefficients $C_{HH'L}$ away from the hydrodynamic linear response regime regime (1.5), e.g. to study $\Delta - \Delta' \gtrsim (\Delta/b_T)^{\frac{1}{d+1}}$.

There are a number of possible interesting extensions, which we leave for future work. We list a few below:

- It should be possible to extend our results to CFTs with anomalies or non-trivial current algebras by studying hydrodynamics with anomalous Ward identities [113–116].

- We have mostly focused on local operators. Certain nonlocal operators, for example in gauge theory, have signatures in the corresponding hydrodynamic theories as higher-form charges [117, 118].

- Operators that are odd under parity (or inversion) can be considered as well, with hydrodynamic tails that depend non-trivially on dimensionality. One can also study heavy operators in CFTs without inversion symmetry, using parity-violating hydrodynamics e.g. in 2+1d [119].

- Boost symmetry plays only a minor role in Sec. 2 – hydrodynamic tails control late time correlators in non Lorentz-invariant QFTs as well. The CFT implications in Sec. 3 rely on a state-operator map. We expect similar results to exist in non-relativistic CFTs (with Schrödinger symmetry), since these also enjoy an operator-state correspondence [120]. The large charge bootstrap has already been extended in this direction [121, 122].

The present work revealed hydrodynamic constraints on CFTs. It is our hope that the favor may one day be returned, with techniques such as crossing and unitarity leading to constraints on dynamics in thermalizing CFTs, e.g. in the form of bounds on transport and thermalization [17, 23, 68, 123, 124].

It would also be interesting to explore if the novel features in late time thermal correlators discussed here have implications for cosmology, where thermal physics enters both in the thermal desription of de Sitter space and through the actual temperature of the universe.

We end with an amusing observation: since reflection positive Euclidean CFTs can be continued to a unitary Lorentzian CFTs [125, 126], the equilibrium properties of certain statistical mechanical systems at their critical point know about hydrodynamics in one lower dimension![†]

# Acknowledgements

I am thankful for inspiring discussions with Nima Afkhami-Jeddi, Alex Belin, Clay Córdova, Gabriel Cuomo, Angelo Esposito, Hrant Gharibyan, Paolo Glorioso, Blaise Goutéraux, Sean Hartnoll, Nabil Iqbal, Kristan Jensen, Steve Kivelson, Umang Mehta, Sasha Monin, Baur Mukhametzhanov, João Penedones, Riccardo Rattazzi, Dam T. Son, and Paul Wiegmann. I also thank Gabriel Cuomo, Blaise Goutéraux and Diego Hofman for useful comments on a

---

[†]This should not be confused with dynamical properties of the fixed point, which are controlled by hydrodynamics in the same amount of spatial dimensions [83].

draft of this paper. This work was supported by the Swiss National Science Foundation and the Robert R. McCormick Postdoctoral Fellowship of the Enrico Fermi Institute.

# A  Detailed hydrodynamic correlators

Correlators involving many indices can be treated by using an index free notation (see e.g. [67]). Consider a spin-$\ell$ operator $\mathcal{O}_{\mu_1 \cdots \mu_\ell}$; its elements involving $\bar{\ell}$ spatial components can be packaged as

$$\mathcal{O}_{(\bar{\ell},\ell)} \equiv z^{i_1} \cdots z^{i_{\bar{\ell}}} \mathcal{O}_{i_1 \cdots i_{\bar{\ell}} \underbrace{0 \cdots 0}_{\ell - \bar{\ell}}}, \tag{A.1}$$

where $z$ lives in $d$-dimensional space (not $d+1$-dimensional spacetime), and satisfies $z^2 = 0$. This projects on the spatially traceless part (spatial traces are related to components with more time indices since $\eta^{\mu_1 \mu_2} \mathcal{O}_{\mu_1 \mu_2 \cdots \mu_\ell} = 0$). We are interested in thermal 2-point functions

$$\langle \mathcal{O}_{(\bar{\ell},\ell)} \mathcal{O}'_{(\bar{\ell}',\ell')} \rangle_\beta (\omega, k) \tag{A.2}$$

in the hydrodynamic regime $\omega \tau_{\text{th}}, k \ell_{\text{th}} \ll 1$.

## A.1  Hydrodynamic loop computation

When $k = 0$, we found in Sec. 2 that correlators (A.2) are dominated by a hydrodynamic loop. For $\bar{\ell}$ even this comes from the following term in the constitutive relation (2.14)

$$\mathcal{O}_{(\bar{\ell},\ell)} = \frac{\lambda_{\bar{\ell}-2} \beta^{\bar{\ell}}}{s^2} (z^i T_{0i}) \partial^{\bar{\ell}-2} (z^j T_{0j}) + \cdots, \tag{A.3}$$

where $\partial \equiv z^i \partial_i$. Note that it does not matter where the derivatives $\partial^{\bar{\ell}-2}$ act, since the $k = 0$ operator $\mathcal{O}_{(\bar{\ell},\ell)}$ is integrated over space. The present normalization defines the dimensionless coefficient $\lambda_{\bar{\ell}-2}$. The contribution of this term to the two-point function between two such operators can be found by factorizing the stress-tensors and using the hydrodynamic correlator (2.7)

$$\langle \mathcal{O}_{(\bar{\ell},\ell)} \mathcal{O}'_{(\bar{\ell}',\ell')} \rangle (t, k = 0) = \frac{\lambda_{\bar{\ell}-2} \lambda'_{\bar{\ell}'-2} \beta^{\bar{\ell}+\bar{\ell}'-4}}{s^2} \times$$
$$2 \int_q (q \cdot z)^{\bar{\ell}-2} (q \cdot z')^{\bar{\ell}'-2} \left[ \frac{(q \cdot z)(q \cdot z')}{q^2} \cos(c_s |q||t|) e^{-\frac{1}{2}\Gamma_s q^2 |t|} \right.$$
$$\left. + \left( z \cdot z' - \frac{(q \cdot z)(q \cdot z')}{q^2} \right) e^{-Dq^2 |t|} \right]^2, \tag{A.4}$$

with $\int_q \equiv \int \frac{d^d q}{(2\pi)^q}$. Terms in the integrand that oscillate with $q$ lead to exponentially decaying terms $\sim e^{-|t|/\tau_{\text{th}}}$ (see e.g. [24]). Dropping these gives

$$\langle \mathcal{O}_{(\bar{\ell},\ell)} \mathcal{O}'_{(\bar{\ell}',\ell')} \rangle (t, k = 0) = \frac{\lambda_{\bar{\ell}-2} \lambda'_{\bar{\ell}'-2} \beta^{\bar{\ell}+\bar{\ell}'-4}}{s^2} \times$$
$$2 \int_q (q \cdot z)^{\bar{\ell}-2} (q \cdot z')^{\bar{\ell}'-2} \left[ \frac{1}{2} \left( \frac{(q \cdot z)(q \cdot z')}{q^2} \right)^2 e^{-\Gamma_s q^2 |t|} + \left( z \cdot z' - \frac{(q \cdot z)(q \cdot z')}{q^2} \right)^2 e^{-2Dq^2 |t|} \right]^2. \tag{A.5}$$

These integrals can be evaluated by noting that when $z^2 = z'^2 = 0$ (for $d > 1$)

$$\int_q \frac{(q \cdot z)^n (q \cdot z')^{n'}}{q^{2m}} e^{-\frac{1}{2}\alpha q^2} = \frac{\delta_{nn'}(z \cdot z')^n}{(2\pi)^{d/2}\alpha^{\frac{d}{2}+n-m}} \frac{n!\Gamma(\frac{d}{2}+n-m)}{2^m\Gamma(\frac{d}{2}+n)} \equiv \frac{\delta_{nn'}(z \cdot z')^n}{(2\pi)^{d/2}\alpha^{\frac{d}{2}+n-m}} I_m^n, \quad (A.6)$$

where in the last step we defined $I_m^n = \frac{n!\Gamma(\frac{d}{2}+n-m)}{2^m\Gamma(\frac{d}{2}+n)}$. One finds

$$\langle \mathcal{O}_{(\bar{\ell},\ell)} \mathcal{O}'_{(\bar{\ell}',\ell')}\rangle(t,k=0) \simeq \delta_{\bar{\ell}\bar{\ell}'}(z \cdot z')^{\bar{\ell}} \frac{\lambda_{\bar{\ell}-2}\lambda'_{\bar{\ell}-2}\beta^{2\bar{\ell}-4}}{(2\pi)^{d/2}s^2}\left[\frac{I_2^{\bar{\ell}}}{(2\Gamma_s|t|)^{\frac{d}{2}+\bar{\ell}-2}} + \frac{2I_2^{\bar{\ell}} - 4I_1^{\bar{\ell}-1} + 2I_0^{\bar{\ell}-2}}{(4D|t|)^{\frac{d}{2}+\bar{\ell}-2}}\right].$$
$$(A.7)$$

Comparison with (2.16) fixes the numerical coefficients there as

$$a_1 = \frac{I_2^{\bar{\ell}}}{(2\pi)^{d/2}} = \frac{\bar{\ell}!/(2\pi)^{d/2}}{\left(\frac{d}{2}+\bar{\ell}-1\right)\left(\frac{d}{2}+\bar{\ell}-2\right)}, \qquad \frac{a_2}{a_1} = \frac{2I_2^{\bar{\ell}} - 4I_1^{\bar{\ell}-1} + 2I_0^{\bar{\ell}-2}}{I_2^{\bar{\ell}}} = 8(\tfrac{d}{2}+\bar{\ell}-2)^2 + 2.$$
$$(A.8)$$

At finite wavevector $k \neq 0$, we found in Sec. 2 that tree-level hydrodynamic contributions dominated the correlation function. Fourier transforming (A.7), collecting these contributions the final answer reads

$$\langle \mathcal{O}_{(\bar{\ell},\ell)} \mathcal{O}_{(\bar{\ell},\ell)}\rangle_\beta(\omega,k) \simeq \frac{2\beta^d}{s_o}(\lambda_{\bar{\ell}-1})^2\left(z \cdot z' - \frac{(k \cdot z)(k \cdot z')}{k^2}\right)\frac{Dk^2(\beta k \cdot z)^{\bar{\ell}-1}(\beta k \cdot z')^{\bar{\ell}-1}}{\omega^2 + (Dk^2)^2}$$
$$+ \frac{2\beta^d}{s_o}\left(\lambda_{\bar{\ell}-1} + \frac{\lambda_{\bar{\ell}}\beta c_s^2 k^2}{\omega}\right)^2\frac{\frac{1}{\beta^2}\Gamma_s\omega^2(\beta k \cdot z)^{\bar{\ell}}(\beta k \cdot z')^{\bar{\ell}}}{(\omega^2 - c_s^2 k^2)^2 + (\Gamma_s\omega k^2)^2} \qquad (A.9)$$
$$+ \frac{\beta^d}{s_o^2}(\lambda_{\bar{\ell}-2})^2\frac{(z \cdot z')^{\bar{\ell}}}{\omega}\left[\left(\frac{\omega\beta^2}{2\Gamma_s}\right)^{\frac{d}{2}+\bar{\ell}-2} + \frac{a_2}{a_1}\left(\frac{\omega\beta^2}{4D}\right)^{\frac{d}{2}+\bar{\ell}-2}\right],$$

where a numerical factor was absorbed in $\lambda_{\bar{\ell}-2}$. Although we have focused on the leading contributions to the correlator in the hydrodynamic regime $\omega\tau_{\text{th}}, k\ell_{\text{th}} \ll 1$, not all possible tensor structures have been 'activated'. Other tensor structures as in Eq. (3.25) will be sensitive to subleading hydrodynamic tails – these will not be computed here.

## A.2 Results for all spin

The hydrodynamic correlators obtained in Sec. 2 hold for any even spin $\ell$ operator with an even number $\bar{\ell}$ of spatial components. In this section, we extend these results to odd $\ell$ and $\bar{\ell}$.

### A.2.1 Odd spin $\ell$

Operators with odd spin $\ell$ still can still decay into hydrodynamic excitations. They also satisfy a constitutive relation of the form (2.10), except that the zero-derivative term $\lambda_0$ is forbidden by CPT[†]. Higher derivative terms in constitutive relations are also constrained by CPT (see e.g. [37]), however these constraints allow for all the $\lambda_i$ in (2.10) as long as $\mathcal{O}$ is not itself a conserved current.

The result (2.15) (or more precisely (A.9)) therefore holds for $\ell$ odd and $\bar{\ell}$ even as long as $\lambda_0$ is not involved, i.e. it holds for $\bar{\ell} \geq 4$ even. Where $\lambda_0$ is involved, it is replaced by a

---

[†]This can also be understood in Euclidean space: $\lambda_0$ gives an equilibrium thermal one-point function which is odd under $\pi$-rotation of the thermal cylinder for odd spin, and must hence vanish, see e.g. [18].

subleading hydrodynamic tail. We detail here the cases $\bar{\ell} = 0, 1, 2$ ($\bar{\ell} \geq 3$ odd will be treated later).

When $\bar{\ell} = 2$, the first two lines in (A.9) are unchanged since they do not involve $\lambda_0$. The second line came from a hydrodynamic loop through $\lambda_0$, the most relevant term when $k = 0$ in the constitutive relation is now

$$\mathcal{O}_{\mu_1 \cdots \mu_\ell} \sim \lambda_1' u_{\mu_1} \cdots u_{\mu_{\ell-1}} \partial_{\mu_\ell} \beta \quad \Rightarrow \quad \mathcal{O}_{(\bar{\ell}=2,\ell)} \sim T_{0i} \partial_j T_{00}, \tag{A.10}$$

and scales as $k(k^{d/2})^2 = k^{d+1}$ (which is indeed less relevant than the forbidden $\lambda_0$ term $\sim k^{d+\bar{\ell}-2} = k^d$). This leads to a long-time tail contribution to the correlator

$$(\ell \text{ odd}) \qquad \langle \mathcal{O}_{(2,\ell)} \mathcal{O}_{(2,\ell)} \rangle(t, k = 0) \sim \frac{1}{t^{\frac{d}{2}+1}}. \tag{A.11}$$

Fourier transforming gives a result similar to the last line of (A.9), replacing the exponent $\frac{d}{2} + \bar{\ell} - 2 \to \frac{d}{2} + 1$.

When $\bar{\ell} = 0$ or 1, the prediction for the correlator $\langle \mathcal{O}_{(\bar{\ell},\ell)} \mathcal{O}_{(\bar{\ell},\ell)} \rangle$ in Eq. (2.13) involves $\lambda_0$. This contribution will be replaced again by less relevant terms e.g. $\mathcal{O}_{(0,\ell)} \sim \partial_i^2 T_{00}$. The resulting correlator will still take the form (2.13), but with additional $\omega$ or $k$ suppression.

## A.2.2 Odd number of spatial indices $\bar{\ell}$

For components of spin-$\ell$ operators with an odd number $\bar{\ell} \geq 3$ of spatial indices, the term $\lambda_{\bar{\ell}-2}$ in (2.14) is a total derivative, and hence will no longer give the dominant contribution to (2.15) when $k \to 0$, which will now come from subleading terms in the constitutive relation

$$
\begin{aligned}
\mathcal{O}_{\langle i_1 \cdots i_{\bar{\ell}} \rangle 0 \cdots 0} = {} & \lambda_{\bar{\ell}} \partial_{i_1} \cdots \partial_{i_{\bar{\ell}}} \delta T_{00} + \lambda_{\bar{\ell}-1} \partial_{i_1} \cdots \partial_{i_{\bar{\ell}-1}} T_{0i_{\bar{\ell}}} \\
& + \lambda'_{\bar{\ell}-1} T_{0i_1} \partial_{i_2} \cdots \partial_{i_{\bar{\ell}}} \delta T_{00} \\
& + \lambda_{\bar{\ell}-3} T_{0i_1} T_{0i_2} \partial_{i_3} \cdots \partial_{i_{\bar{\ell}-1}} T_{0i_{\bar{\ell}}} + \lambda'_{\bar{\ell}-2} T_{0i_1} T_{0i_2} \partial_{i_3} \cdots \partial_{i_{\bar{\ell}}} \delta T_{00} + \cdots,
\end{aligned}
\tag{A.12}
$$

where the $\lambda'$ terms come from different distributions of derivatives in (2.10). The first line gives tree-level contributions to $\langle \mathcal{O} \mathcal{O} \rangle$, the second line gives 1-loop contributions, and so on. Since the loop contributions will only dominate terms in the first line when $k \to 0$, we have dropped total derivative terms such as $\lambda_{\bar{\ell}-2}$. The two most relevant loop contributions come from $\lambda'_{\bar{\ell}-1}$ (1-loop) and $\lambda_{\bar{\ell}-3}$ (2-loop). These terms scale as

$$\lambda'_{\bar{\ell}-1} : \ k^{\bar{\ell}-1+d}, \qquad \lambda_{\bar{\ell}-3} : \ k^{\bar{\ell}-3+\frac{3d}{2}}, \tag{A.13}$$

so that $\lambda_{\bar{\ell}-3}$ dominates for spatial dimensions $d \leq 4$ (and $\lambda'_{\bar{\ell}-1}$ dominates in higher dimensions). The leading correlator then behaves as

$$\langle \mathcal{O}_{(\bar{\ell},\ell)} \mathcal{O}_{(\bar{\ell},\ell)} \rangle(t, k = 0) \sim \begin{cases} \dfrac{(\lambda_{\bar{\ell}-3})^2}{t^{d+\bar{\ell}-3}} & \text{when } d \leq 4, \\[3mm] \dfrac{(\lambda'_{\bar{\ell}-1})^2}{t^{\frac{d}{2}+\bar{\ell}-1}} & \text{when } d > 4. \end{cases} \tag{A.14}$$

One final special case is when $\bar{\ell} = \ell = 3$. Then as shown in A.2.1, $\lambda_{\bar{\ell}-3} = \lambda_0$ is forbidden by CPT, so that the top line is replaced by a subleading hydrodynamic tail.

## A.3 Subleading tails

In Sec.2, the leading hydrodynamic contribution to correlators $\langle \mathcal{O}_{(\bar{\ell},\ell)} \mathcal{O}_{(\bar{\ell},\ell)} \rangle(\omega, k)$ were found by matching the operators $\mathcal{O}_{(\bar{\ell},\ell)}$ to composite hydrodynamic operators as in Fig. 1, and then Gaussian factorizing the hydrodynamic fields. Gaussian factorization however only holds at the lowest energies (or smallest $\omega, k$) and irrelevant interactions give subleading corrections to the correlators. These corrections can be captured systematically by using a dissipative effective field theory for fluctuating hydrodynamics, see e.g. [25, 31, 36]. This was performed to next to leading order in [42] for simple diffusion.

In this appendix, we will illustrate the structure of these subleading corrections with a specific example. We will do so without using the full effective action, and therefore miss certain subleading contributions to the correlator. However, important qualitative features of the answer (such as the analytic structure) will be captured.

Consider the component of a spin-$\ell$ operator with only time indices $\mathcal{O}_{(0,\ell)} \equiv \mathcal{O}_{0\cdots 0}$. Its constitutive relation (2.10) can be expanded using (2.5)

$$
\begin{aligned}
\mathcal{O}_{(0,\ell)} &= \lambda_0(\beta) + \cdots \\
&= \lambda_0 - \partial_\beta \lambda_0 \frac{\beta^2 c_s^2}{s} \delta T_{00} + \frac{1}{2}(\partial_\beta^2 \lambda_0) \left( \frac{\beta^2 c_s^2}{s} \right)^2 (\delta T_{00})^2 + \cdots,
\end{aligned}
\tag{A.15}
$$

where $\cdots$ denotes higher derivative terms that we will ignore. The two point function $\langle \mathcal{O}_{(\bar{\ell},\ell)} \mathcal{O}_{(\bar{\ell},\ell)} \rangle$ will receive a single tree-level contribution from the linear term $\langle \delta T_{00} \delta T_{00} \rangle$, which is given in (2.13a). It will receive 1-loop corrections from $\langle \delta T_{00} \delta T_{00} \rangle$, $\langle \delta T_{00} \delta T_{00}^2 \rangle$ and $\langle \delta T_{00}^2 \delta T_{00}^2 \rangle$. The first two come from interactions in the action and will not be captured here – we will focus on the last. At leading order it can be factorized

$$
\langle \delta T_{00}^2 \delta T_{00}^2 \rangle(\omega, k) \simeq 2 \int d^d x \, dt \, e^{ik \cdot x - i\omega t} \left( \langle T_{00} T_{00} \rangle(x, t) \right)^2.
\tag{A.16}
$$

The hydrodynamic correlator appearing in the integrand can be obtained from (2.7)

$$
\langle T_{00} T_{00} \rangle(x, t) \simeq \frac{s}{2\beta^2 c^2} \frac{e^{-(x+c|t|)^2/2\Gamma_s|t|} + e^{-(x-c|t|)^2/2\Gamma_s|t|}}{(2\pi\Gamma_s|t|)^{d/2}},
\tag{A.17}
$$

so that performing the integral yields

$$
\langle \delta T_{00}^2 \delta T_{00}^2 \rangle(\omega, k) \simeq \frac{s^2/(\beta c_s)^4}{2(4\pi\Gamma_s)^{d/2}} \left[ \frac{\left(-i(\omega - c_s k) + \frac{1}{4}\Gamma_s k^2\right)^{\frac{d-2}{2}}}{\Gamma(1 - \frac{d}{2})} + \frac{\left(-i(\omega + c_s k) + \frac{1}{4}\Gamma_s k^2\right)^{\frac{d-2}{2}}}{\Gamma(1 - \frac{d}{2})} \right].
\tag{A.18}
$$

The quantity in square brackets is formally divergent – the divergence can be treated in dimensional regularization by expanding around integer $d$ and throwing away the divergent piece (this UV divergence can be absorbed in the bare transport parameters [42]). This gives

$$
\frac{A^{\frac{d-2}{2}}}{\Gamma(1 - \frac{d}{2})} \rightarrow \frac{(-1)^{\lfloor \frac{d}{2} \rfloor} \pi A^{\frac{d-2}{2}}}{\Gamma(\frac{d}{2})} \cdot
\begin{cases}
1 & \text{for } d \text{ even}, \\
\frac{1}{\pi} \log A & \text{for } d \text{ even},
\end{cases}
\tag{A.19}
$$

with $A = -i(\omega \pm c_s k) + \frac{1}{4}\Gamma_s k^2$. Eq. (A.18) features a branch cut, with branch point at

$$
\omega = \pm c_s k - \frac{i}{4}\Gamma_s k^2,
\tag{A.20}
$$

which should be contrasted to the pole in the tree-level part of the correlator (2.13a), at $\omega = \pm c_s k - \frac{i}{2}\Gamma_s k^2$. This is the sound analog of the two-diffuson branch cut at $\omega = -\frac{i}{2}Dk^2$ found in [42]. The analytic structure of hydrodynamic correlators is shown in Fig. 4. For generic $\omega \sim k$, (A.18) is suppressed compared to the tree-level contribution (2.13a) – however it dominates as $k \to 0$ where one finds $\langle \mathcal{O}_{(0,\ell)}\mathcal{O}_{(0,\ell)}\rangle(\omega, k=0) \sim (\partial_\beta^2 \lambda_0)^2 \omega^{\frac{d}{2}-1}$.

Higher-loop corrections will lead to additional branch points at the threshold for production of $n$ diffusons $\omega = -\frac{i}{n}Dk^2$ or $n$ sound modes $\omega = \pm c_s k - \frac{i}{2n}\Gamma_s k^2$, but the discontinuities across the cuts are increasingly suppressed at small $\omega$ and $k$. However, the Fourier transform $\omega \to t$ picks up these non-analyticities, and the leading behavior of $G(t, k)$ is controlled by multi-diffuson decay at late time, as shown in Sec. 2.3.

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
