# Peer review of "Heavy Operators and Hydrodynamic Tails"

_SciPost Physics, doi:SciPost Phys. 9, 034 (2020)_

## Round 1 · Referee Report · Raghu Mahajan (Referee 1) · 2020-8-9

Strengths

1- Multiple physically important results.

Weaknesses

No significant weakness.

Report

I recommend publication, but also recommend to make the following clarificational changes.

Requested changes

1- Just before equation (1.3), it is not clear what the phrase "this hydrodynamic correlator" is referring to. Is it referring to the one-point function, or some hydrodynamic two-point function?

2- After equation (1.3), the sentence "The singular dependence on \Delta - \Delta' and J-J' featuring the hydrodynamic sound pole" is a bit unclear. It would be nice to have one sentence saying where exactly the singularity is in terms of Delta and J, and why this is related to the physics of sound.

3- The second term in (1.5) should also have an \bar{\ell} instead of \ell?

4- The caption of figure 1 is a little confusing. "Neutral operators in finite temperature QFT are ‘light’ as they can decay into long-lived hydrodynamic excitations". One would think that the notion of lightness is just set by the dimension of the operator, and does not have anything to do with whether or not it can so decay.

5- Before equation (1.7) we again have the phrase "This hydrodynamic correlator" without referring to which correlator. It would be helpful to the reader to point to some equation or say "two-point function".

6- I presume in "averaging over 10 operators", one can replace 10 by any number A with an error 1/\sqrt{A}? It is pretty random that the ten comes out of nowhere.

7- Is (1.7) supposed to match on to (1.3) as one increases J-J'? From the formulas it doesn't seem like the exponents in (1.7) depend on d or \bar{\ell}. It would be nice to clarify this point.

8- In the footnote on page 9 "Neither fixed point describes CFTs in d = 1, where the enhanced symmetries completely fix finite temperature physics". This statement is true if the spatial manifold is the real line. If the spatial manifold is a circle, the torus one-point function even in a CFT_2 are complicated objects.

9- Typo on page 11, "the stress stress tensor".

10- "Although so acts as a loop counting parameter in fluctuating hydrodynamics, we emphasize that the perturbative expansion is controlled even when s_0 \sim 1 because hydrodynamic interactions are irrelevant." This comment is a bit mysterious to the referee. Don't we need s_0 large just in order to be able to talk about hydrodynamics? If we have a very dilute system of water molecules, it probably isn't described by hydrodynamics. Of course, in general one would need to replace s\beta^d by s (mfp)^d. In any case, it would be helpful to explain this point a bit more from the perspective of the author.

11- It would be very helpful to give a few more lines of algebra around equation 2.29. For example, if we take the derivative of 2.28 wrt n to determine the value of n for which 2.28 is largest, we get

0 = log n - d log[ 1/(k \ell_th) ] + D k^2 t/n^2.

Now equation 2.29 is balancing the last two terms, but why do we get to ignore the log n term? Of course, what is happening is that the n! hasn't kicked in during regime 3, and being able to ignore the log n above is why we get the upper limit on regime 3. It would be nice to add these small algebraic details for the convenience of the reader.

12- How did the \bar{\ell} appear in 2.30? It is not there in 2.28.

13- It is not clear what the 'weight m' means in equation 3.7. Is it the scaling dimension of the state labelled by H? Or are these the weights in the sense of representation theory of the rotation group?

  • validity: top
  • significance: top
  • originality: top
  • clarity: high
  • formatting: perfect
  • grammar: perfect

Author:  Luca Delacrétaz  on 2020-08-13  [id 926]

(in reply to Report 1 by Raghu Mahajan on 2020-08-09)
Category:
remark
answer to question

Dear Raghu,

Thank you for your detailed report. A full list of changes following your suggestions can be found in the new submission; below I address several of your questions more specifically:

1,2- The correlator in question is the thermal two-point function, which is now shown in an extra equation (1.3), clarifying also why the corresponding pole in the OPE is related to hydrodynamic sound.

7- The expressions in Eq. (1.8) (previously (1.7)) and (1.4) (previously (1.3)) come from different contributions to the thermal two-point function: (1.8) is one-loop and hence subleading compared to (1.4) in the hydrodynamic regime (1.5). However (1.4) vanishes when J=J', so that (1.8) is the leading result in that case. I have added a comment below (1.8) clarifying this point, and with a reference to the general result (3.27) where both contributions enter.

10 - Any interacting system, even very dilute or weakly interacting, will have a hydrodynamic description at finite temperature. Of course the thermalization length (or mean-free path) may be prohibitively large in practice, e.g. larger than system size. In the present context of describing heavy operators in CFTs, there will be a parametric window with hydrodynamic behavior as long as one studies heavy enough operators, see comment below Eq. (1.5), and the explicit example of weakly coupled superfluids in Eq. (4.30).

Now the "loop counting parameter" $s_o$ measures the strength of hydrodynamic loop corrections, rather than how strongly coupled the original QFT is. While it is true that water has a large loop counting parameter (the loop counting parameter there is slightly different from $s_o$ since there is an extra number conservation), and fluctuation effects are therefore further suppressed, the $s_o$ of the quark-gluon plasma ($s_o$~10) or spin chains is not parametrically large. Even in these situations, the hydrodynamic loop expansion is controlled because interactions are irrelevant. I therefore expect interacting QFTs with $s_o\sim 1$ (e.g. the 3d Ising model) to also be described by fluctuating hydrodynamics at finite temperature.

13- The weight $m$ here refers to the representation of the Lorentz group; I have added clarifying comments.

Thank you, Luca

---

## Round 2 · List of Changes

List of changes (numbers refer to comments from the referee):

1,2- Added Eq. (1.3) with thermal two-point function, and discussion above on the location of the sound pole.

4- Changed 'light' to 'long-lived'.

5- Changed 'correlator' to 'two-point function' and referred to the appropriate equation.

6- Replaced "10" with "n".

7- Extra comment below (1.8) explaining how the general result involves both (1.4) and (1.8).

8- Corrected footnote on p.9.

10- Added a footnote on p.14 with the value of the dimensionless entropy s_o for the quark-gluon plasma.

11- Added explanation for the derivation of (2.29).

12- The \bar{\ell} was removed from Eq. (2.30).

13- Added definition of J and m above (3.7).

Other changes:

power of k\ell_th in (2.30) and (2.31)

power of c_1 in (4.22) and (4.24)

---

## Editorial Decision

published